# Asymmetric Dimethylaminohydrolase Gene Polymorphisms Associated with Preeclampsia Comorbid with HIV Infection in Pregnant Women of African Ancestry

**DOI:** 10.3390/ijms26073271

**Published:** 2025-04-01

**Authors:** Mbuso Herald Mthembu, Samukelisiwe Sibiya, Zinhle Pretty Mlambo, Nompumelelo P. Mkhwanazi, Thajasvarie Naicker

**Affiliations:** 1Department of Obstetrics and Gynaecology, Nelson R. Mandela School of Medicine, University of KwaZulu-Natal, Durban 4041, South Africa; 214556782@stu.ukzn.ac.za; 2Optics and Imaging Centre, Doris Duke Medical Research Institute, College of Health Sciences, University of KwaZulu-Natal, Durban 4041, South Africa; 214513843@stu.ukzn.ac.za; 3HIV Pathogenesis Programme, Doris Duke Medical Research Institute, College of Health Sciences, University of KwaZulu-Natal, Durban 4041, South Africa; 215013361@stu.ukzn.ac.za (S.S.); mkhwanazi@ukzn.ac.za (N.P.M.)

**Keywords:** preeclampsia, ADMA, DDAH, endothelial impairment, HIV infection, single-nucleotide polymorphisms

## Abstract

Asymmetric dimethylarginine (ADMA) is an endogenous nitric oxide synthase (NOS) inhibitor associated with vascular disease, which is prevalent in human plasma. Two isoforms of the enzyme dimethylarginine dimethylaminohydrolase (DDAH), DDAH 1 and 2, degrade ADMA. This study investigates the association of *DDAH 1* (*rs669173*, *rs7521189*) and *DDAH 2* gene polymorphisms (*rs805305*, *rs3131383*) with the risk of preeclampsia (PE) comorbidity with human immunodeficiency virus (HIV) infection in pregnant women of African ancestry. A total of 405 women were enrolled in this study: 204 were PE, 201 were normotensive pregnant, and 202 were HIV positive. DNA was extracted from whole blood, and SNPs (*rs669173*, *rs7521189*, *rs805305*, and *rs3131383*) were amplified to detect single-nucleotide polymorphisms (SNPs). After PCR amplification, allelic discrimination was examined. Comparisons were conducted utilizing the Chi-squared test. Our findings indicated that preeclamptic women displayed a greater prevalence of the three variants compared to those with both PE and HIV infection. There is an association between the *rs669173* and *rs7521189* SNPs of the *DDAH 1* gene and *rs3131383* of the *DDAH 2* gene, which could play a role in reducing the bioavailability of nitric oxide (NO), which affects endothelial function, leading to the development of PE in pregnant women of African ancestry. In contrast, the *rs805305* variant of the *DDAH 2* gene was not significantly associated with PE development. Interestingly, none of the SNPs investigated correlated with HIV infection or could be attributed to the human allelic variant influence on HIV infection outcome.

## 1. Introduction

Preeclampsia (PE) is a serious pregnancy-related disorder characterized by hypertension with/without proteinuria and multi-organ dysfunction that leads to adverse maternal and fetal outcomes [1,2,3]. PE affects approximately 2–8% of pregnancies worldwide, particularly in developing countries [4,5]. The World Health Organization reports that 16% of maternal fatalities that occur within sub-Saharan Africa are due to hypertensive disease in pregnancy, primarily PE and eclampsia [6,7]. The etiology of PE remains unclear. PE is known as a multi-factorial condition, with its pathophysiology not ascribable to a singular component. Instead, it arises from a complex interplay of genetic, immunogenic, and environmental influences [8,9]. Single-nucleotide polymorphisms (SNPs) are among the many genetic and environmental variables contributing to the pathophysiology of PE [9]. SNPs are naturally occurring variants in the human genome, characterized as genomic loci where two or more alternative bases are present with significant frequency [10,11]. It is widely accepted that PE is associated with a hypo-perfused placenta that emanates from a shallow cytotrophoblast migration and an absence of spiral artery remodeling within the myometrium [12,13]. The resultant reduction in artery diameter predisposes a deficient blood supply, with consequent ischemia that precipitates certain factors’ release into circulation, exacerbating the widespread pathognomonic endothelial injury [12,14,15,16,17,18]. Endothelial dysfunction results in vasoconstriction and decreased blood flow to several essential organs [14,19,20,21,22]. Reduced nitric oxide (NO) production and sensitivity are signs of endothelial dysfunction, pre-empting a vascular imbalance that results in a pro-inflammatory, pro-thrombotic, and less compliant vessel wall [23].

Asymmetric dimethylarginine (ADMA) is an endogenous NO synthase (NOS) inhibitor, which plays a crucial role in regulating vascular tone and endothelial function [24]. ADMA reduces NO production, consequently leading to endothelial dysfunction and cardiovascular events [25]. It is derived from the catabolism of proteins containing methylated arginine residues released from protein hydrolysis within the nucleus and is implicated in RNA processing and transcriptional control [26]. ADMA concentration in plasma and tissue is controlled by one of two isoforms of dimethylarginine dimethylaminohydrolase 1 and 2 (DDAH 1 and 2), responsible for over 70% of its breakdown [27,28,29,30]. DDAH 1, a major enzyme, is highly expressed in the liver and kidneys, which are the main sites of NOS [31,32,33]. Studies have shown that DDAH 2 dominates within tissues and the lungs, where it controls NO-mediated vasodilation [27,32]. Genes encoding enzymes involved in ADMA production and metabolism may undergo SNPs in the hypoxic microenvironment of PE [34]. Moreover, therapeutic interventions aimed at DDAH activity are proposed to enhance vascular health [35].

Human immunodeficiency virus (HIV) proteins (Tat, Nef, and gp120) impair immune and endothelial function [36]. Plasma ADMA levels are elevated in HIV-1 individuals, correlating with the virus-induced immune activation and increased levels of methylated arginine species [37]. Antiretroviral therapy (ART) also affects angiogenic homeostasis and thus influences endothelial function [38,39]. A strong association of SNPs in the *DDAH 1* and *DDAH 2* genes and ADMA levels in patients with diabetes mellitus has been reported; however, limited information exists on their particular influence on PE development and the comorbidity with HIV in women of African descent [40]. Understanding these genetic variations may provide valuable insights into underlying mechanisms and potential therapeutic targets in women of African ancestry.

Despite the high prevalence of both PE (14.8%) and HIV infection (13.9%) in South Africa, there is limited research investigating the genotypic and allelic variation of *DDAH* genes in pregnant women of African ancestry. We hypothesize that mutations in the *DDAH* gene are strongly correlated with an elevated risk of PE in pregnant women of African descent who are also co-infected with HIV. Therefore, this study aims to investigate the association of *DDAH 1* (*rs669173*, *rs7521189*) and *DDAH 2* genes (*rs805305*, *rs3131383*) in PE comorbid with HIV infection. To our knowledge, this is the first study to investigate genetic variation in DDAH enzymes and their association with ADMA levels in this duality in women of African ancestry. These polymorphisms may exacerbate endothelial dysfunction and impair NO synthesis, intensifying the severity of PE within this population.

## 2. Results

### 2.1. Clinical Characteristics

The study included 405 pregnant women, comprising 204 with PE (50.37%) and 201 normotensive individuals (49.63%). The PE group was categorized by gestational age into early onset preeclampsia (EOPE) (*n* = 102; 25.19%) and late onset preeclampsia (LOPE) (*n* = 102; 25.19%). Additionally, it was stratified by HIV status into EOPE^−^ (*n* = 50; 12.35%), EOPE^+^ (*n* = 52; 12.84%), LOPE^−^ (*n* = 51; 12.59%), and LOPE^+^ (n = 51; 12.59%) groups. The normotensive cohort was categorized by HIV status into N^−^ (*n* = 102; 25.19%) and N^+^ (*n* = 99; 24.44%).

Table 1 presents the demographics and clinical data of the study population. The preeclamptic group exhibited a significantly lower gestational age at delivery compared to the normotensive group (*p* < 0.0001), and their systolic and diastolic blood pressure values were markedly elevated relative to the normotensive group (*p* < 0.0001). Maternal weight exhibited a statistically significant difference between the PE and normotensive groups (*p* = 0.0038), with the PE group demonstrating a notably higher weight (*p* = 0.0005).

There were no significant differences between HIV-negative vs. HIV-positive normotensive and preeclamptic groups. Maternal age was different between the N vs. EOPE (*p* < 0.0001) and the EOPE vs. the LOPE (*p* < 0.0068) groups.

### 2.2. Genotype and Allele Frequencies of SNPs rs669173, rs7521189, rs805305, and rs3131383

Four genetic models were assessed for their correlations with PE associated with HIV infection across the four variations analyzed and is depicted in Table 2, Table 3, Table 4 and Table 5.

A scatter plot (Figure 1) illustrates the distribution of genotypic and allelic frequencies for the SNPs *rs669173*, *rs7521189*, *rs805305*, and *rs3131383* among pregnant women involved in the study cohort, as detailed in Appendix A. The x-axis represents the specific alleles for each SNP, while the y-axis indicates their corresponding frequencies within the study population. Each data point on the scatter plot represents an individual observation and is color-coded according to genotype (e.g., GG, AA, CG, TC, etc.) to facilitate visual differentiation between homozygous and heterozygous variants. The clustering or dispersion of these points highlights patterns of allele distribution, revealing trends in genetic variation within the studied cohort. The accumulation of points in certain regions of the plot suggests a higher prevalence of specific alleles or genotypes within defined subgroups, reflecting potential associations between these genetic variants and clinical or demographic characteristics.

### 2.3. rs669173

#### 2.3.1. Normotensive HIV-Negative Pregnant Women vs. Preeclamptic HIV-Negative Pregnant Women

The genotype frequencies of *rs669173* in the normotensive HIV-negative group were as follows: TT- 40 (39.22%), TC- 45 (41.12%), and CC- 17 (16.67%). In contrast, the preeclamptic HIV-negative group had frequencies of TT- 28 (27.72%), TC- 45 (44.55%), and CC- 28 (27.72%). The allelic frequencies of T and C were 125 (61.27%) and 79 (37.73%) in the normotensive HIV-negative group, compared to 101 (50%) and 101 (50%) in the preeclamptic HIV-negative group. There was a significant difference between the mean frequency of alleles C and T for *rs669173* (*p* = 0.0232) (Appendix A; Figure 1).

The genotypic frequencies of TT vs. CC [OR = 0.4250; 95% CI (0.1963–0.9202); adjusted *p* = 0.0352] indicated significant associations between the normotensive and preeclamptic HIV-negative groups. However, no significant associations were observed for TT vs. TC and TC vs. CC within the same groups. Dominant, recessive, and over-dominant alleles (TT vs. TC + CC; TT + TC vs. CC; TT + CC vs. TC) exhibited no significant associations between normotensive HIV-negative pregnant women and preeclamptic HIV-negative individuals. The allelic frequency association between T and C showed a significant difference between the normotensive HIV-negative and the preeclamptic HIV-negative group [OR = 0.6320; 95% CI (0.4260–0.9375); adjusted *p* = 0.0278] (Table 2).

#### 2.3.2. Normotensive HIV-Positive vs. Preeclamptic HIV-Positive Group

The genotype frequencies of *rs669173* among normotensive HIV-positive pregnant women were TT- 33 (33.33%), TC- 22 (22.22%), and CC- 24 (24.24%). The preeclamptic HIV-positive group showed frequencies of TT- 30 (29.13%), TC- 45 (43.69%), and CC- 28 (27.18%). The allelic frequencies of T and C were 108 (54.55%) and 90 (45.45%) in the normotensive HIV-positive group, compared to 105 (50.97%) and 101 (49.03%) in the preeclamptic HIV-positive group (Appendix A).

The data showed no discernible differences in the genotypic frequencies of TT vs. CC, TT vs. TC, and TT vs. CC between the preeclamptic and normotensive HIV-positive groups. The dominant (TT vs. TC + CC), recessive (TT + TC vs. CC), and over-dominant alleles exhibited no significant correlation between the normotensive HIV-positive group and the preeclamptic HIV-positive group (Table 2).

#### 2.3.3. Normotensive Group vs. Preeclamptic Groups Regardless of HIV Status

The genotype frequencies of *rs669173* in the normotensive group were TT- 73 (36.32%), TC- 87 (43.28%), and CC- 41 (20.40%) compared to TT- 58 (28.43%), TC- 90 (44.12%), and CC- 56 (27.45%) in the preeclamptic group, irrespective of HIV status. The allelic frequencies T and C were 233 (57.96%) and 169 (42.04%) in the normotensive compared to 206 (50.49%) and 202 (49.51%) in the preeclamptic group, regardless of HIV status (Appendix A).

The genotypic frequency relationship of gene polymorphisms co-dominant TT vs. CC, TT vs. TC, and TC vs. CC demonstrated no significant correlation between the normotensive and preeclamptic groups unrelated to HIV status. Dominant (TT vs. TC + CC), recessive (TT + TC vs. CC), and/or over-dominant (TT + CC vs. TC) alleles exhibited no significant correlation between the normotensive and preeclamptic groups. Furthermore, the correlation of the allelic frequency between T and C demonstrated a significant connection between normotensive pregnant women and the preeclamptic cohort [OR = 0.7397; 95% CI (0.5605–0.9761); adjusted *p* = 0.0346] (Table 2).

#### 2.3.4. Early Onset Preeclampsia vs. Late Onset Preeclampsia Groups Irrespective of HIV Status

The frequencies of *rs669173* in the EOPE group were TT- 30 (29.41%), TC- 44 (43.14%), and CC- 28 (27.45%), while in the LOPE group, they were TT- 28 (27.45%), TC- 46 (45.10%), and CC- 28 (27.45%), irrespective of HIV status. The allele frequencies for T and C were 104 (50.98%) and 100 (49.02%) in the EOPE group, compared to 102 (50.00%) for both alleles in the LOPE group, regardless of HIV status (Appendix A).

The genotypic frequency connection of gene polymorphism co-dominant TT vs. CC, TT vs. TC, and TC vs. CC revealed no significant association of EOPE compared to LOPE groups, irrespective of HIV status. Likewise, the genotypic frequency exhibited no significant correlation between the EOPE and LOPE groups. No significant connection was seen between the EOPE and LOPE groups concerning dominant (TT vs. TC + CC), recessive (TT + TC vs. CC), and/or over-dominant (TT + CC vs. TC) alleles (Table 2).

#### 2.3.5. Normotensive Group vs. Early Onset Preeclampsia Group Irrespective of HIV Status 

The *rs669173* genotype frequencies were TT- 73 (36.32%), TC- 87 (43.28%), and CC- 41 (20.40%) in the normotensive group and TT- 30 (29.41%), TC- 44 (43.14%), and CC- 28 (27.45%) in the EOPE group, irrespective of HIV status. The allelic frequencies T and C were 233 (57.96%) and 169 (42.04%) in the normotensive group and 104 (50.98%) and 100 (49.02%) in the EOPE group, irrespective of HIV status (Appendix A).

The genotypic frequency relationship of gene polymorphism co-dominant TT vs. CC, TT vs. TC, and TC vs. CC exhibited no significant correlation across the normotensive and EOPE groups. There was also no correlation between normotensive pregnant women and the EOPE group dominant (TT vs. TC + CC), recessive (TT + TC vs. CC), and over-dominant (TT + CC vs. TC) alleles, regardless of HIV status. The allelic frequency association of T versus C exhibited no significant difference between the normotensive and EOPE groups (Table 2).

#### 2.3.6. Normotensive Group vs. Late Onset Preeclampsia Group Irrespective of HIV Status

The *rs669173* genotype frequencies in the normotensive group were TT- 73 (36.32%), TC- 87 (43.28%), and CC- 41 (20.40%). In contrast, the LOPE group exhibited frequencies of TT- 28 (27.45%), TC- 46 (45.10%), and CC- 28 (27.45%), regardless of HIV status. The allelic frequencies in the normotensive group were T and C at 233 (57.96%) and 169 (42.04%), respectively, compared to 102 (50.00%) for both alleles in the LOPE group, irrespective of HIV status (Appendix A).

The genotypic frequency association of gene polymorphism co-dominant TT vs. CC, TT vs. TC, and TC vs. CC revealed no significant association between normotensive pregnant women and the LOPE group, regardless of HIV status. There was no significant association observed between dominant, recessive, and over-dominant alleles in normotensive individuals compared to the LOPE group [(TT vs. TC + CC); (TT + TC vs. CC); (TT + CC vs. TC)]. Also, the allelic frequency association of T vs. C exhibited no significant difference between normotensive pregnant women and the LOPE group (Table 2).

#### 2.3.7. HIV-Negative vs. HIV-Positive Women Regardless of Pregnancy Type

The *rs669173* genotype frequencies in the HIV-negative group were TT- 68 (33.50%), TC- 90 (44.33%), and CC- 45 (22.17%) compared to the HIV-positive group, TT- 63 (31.19%), TC- 87 (43.07%), and CC- 52 (25.74%), regardless of pregnancy type. The allelic frequencies of T and C were 226 (55.67%) and 180 (44.33%) in the HIV-negative group, respectively, compared to 213 (52.72%) and 191 (47.28%) in the HIV-positive group, irrespective of pregnancy type (Appendix A).

The genotypic frequency connection of gene polymorphism co-dominant TT vs. CC, TT vs. TC, and TC vs. CC demonstrated no significant correlation between HIV-negative and HIV-positive groups. The dominant (TT vs. TC + CC), recessive (TT + TC vs. CC), and over-dominant (TT + CC vs. TC) alleles exhibited no significant correlation between HIV-negative and HIV-positive pregnant women. The allelic frequency association of T vs. C demonstrated no significant difference between the HIV-negative and HIV-positive groups (Table 2).

### 2.4. rs7521189

#### 2.4.1. Normotensive vs. Preeclamptic HIV-Negative Group

The genotype frequencies of *rs7521189* were GG- 28 (27.45%), GA- 42 (41.18%), and AA- 32 (31.37%) in the normotensive HIV-negative group compared to GG- 30 (29.70%), GA- 51 (50.50%), and AA- 20 (19.80%) in the preeclamptic HIV-negative group. The allelic frequencies of G and A were 98 (48.04%) and 106 (51.96%) in the normotensive HIV-negative group compared to 111 (54.95%) and 91 (45.05%) in the preeclamptic HIV-negative group. A notable disparity was observed in the mean frequency of alleles G and A for *rs7521189* (*p* = 0.038) (Appendix A; Figure 1).

The genotypic frequency associations of gene polymorphisms GG vs. AA, GG vs. GA, and GA vs. AA exhibited no significant correlation between normotensive HIV-negative individuals and preeclamptic HIV-negative individuals. Furthermore, examinations of the dominant (GG vs. GA + AA), recessive (GG + GA vs. AA), and over-dominant (GG + AA vs. GA) alleles revealed no significant relationships between the normotensive HIV-negative and preeclamptic HIV-negative groups. The allelic frequency association between G and A exhibited no significant difference between the two groups (Table 3).

#### 2.4.2. Normotensive vs. Preeclamptic HIV-Positive Group

The genotype frequencies of *rs7521189* were GG- 27 (27.27%), GA- 47 (47.47%), and AA- 25 (25.25%) in normotensive HIV-positive pregnant women compared to GG- 37 (35.92%), GA- 48 (46.60%), and AA- 18 (17.48%) in preeclamptic HIV-positive women. The allele frequencies G and A were 101 (51.01%) and 97 (48.99%) in the normotensive HIV-positive group and 122 (59.22%) and 84 (40.78%) in the preeclamptic HIV-positive group (Appendix A).

The genotypic frequency associations of gene polymorphisms GG vs. AA, GG vs. GA, and GA vs. AA revealed no significant correlation between normotensive HIV-positive individuals and preeclamptic HIV-positive individuals. Furthermore, comparisons of the dominant (GG vs. GA + AA), recessive (GG + GA vs. AA), and over-dominant (GG + AA vs. GA) alleles revealed no significant relationships between the normotensive HIV-negative group and the preeclamptic HIV-negative group. The allelic frequency connection between G and A also exhibited no significant difference between the two groups (Table 3).

#### 2.4.3. Normotensive Group vs. Preeclamptic Group Irrespective of HIV Status

The *rs7521189* genotype frequencies in the normotensive group were GG- 55 (27.36%), GA- 89 (44.28%), and AA- 57 (28.36%), but in the preeclamptic group, they were GG- 67 (32.84%), GA- 99 (48.53%), and AA- 38 (18.63%), irrespective of HIV status. The allele frequencies for G and A were 199 (49.50%) and 203 (50.50%) in the normotensive group, respectively, compared to 233 (57.11%) and 175 (42.89%) in the preeclamptic group, regardless of HIV status (Appendix A).

The genotypic frequency connection of the co-dominant gene polymorphism GG vs. AA [OR = 1.827; 95% CI (1.061–3.148); adjusted *p* = 0.0398] demonstrated a strong correlation between normotensive pregnant women and the preeclamptic group, regardless of HIV status. The genotypic frequency of the dominant GG compared to GA + AA [OR = 1.729; 95% CI (1.084–2.759); adjusted *p* = 0.0256] demonstrated a strong relationship between the normotensive and preeclamptic groups, irrespective of HIV status. Additionally, comparisons of the recessive (GG + GA vs. AA) and over-dominant (GG + AA vs. GA) alleles revealed no significant connection between the normotensive and preeclamptic groups, regardless of HIV status. Furthermore, the allelic frequency relationship of G and A exhibited a significant disparity between the normotensive and preeclamptic groups, regardless of HIV status [OR = 1.358; 95% CI (1.030–1.792); adjusted *p* = 0.0346] (Table 3).

#### 2.4.4. Early Onset Preeclampsia vs. Late Onset Preeclampsia Groups Irrespective of HIV Status

The *rs7521189* frequencies in the EOPE group were GG- 30 (29.41%), GA- 52 (50.98%), and AA- 20 (19.61%), compared to GG- 37 (36.27%), GA- 47 (46.08%), and AA- 18 (17.65%) in the LOPE group, irrespective of HIV status. The allele frequencies for G and A were 112 (54.90%) and 92 (45.10%) in the EOPE group, compared to 121 (59.31%) and 83 (40.69%) in the LOPE group, regardless of HIV status (Appendix A). 

The genotypic frequency connection of the co-dominant gene polymorphism GG vs. AA revealed no significant correlation between the EOPE and LOPE groups, regardless of HIV status. The genotypic frequencies of GG vs. GA and GA vs. AA exhibited no significant connection between the EOPE and LOPE groups, regardless of HIV status. Additionally, comparisons of the dominant (GG vs. GA + AA), recessive (GG + GA vs. AA), and over-dominant (GG + AA vs. GA) alleles revealed no significant connection between the EOPE and LOPE groups. The allelic frequency association of G and A exhibited no significant difference in the EOPE compared to the LOPE group, regardless of HIV status (Table 3).

#### 2.4.5. Normotensive Group vs. Early Onset Preeclampsia Group Irrespective of HIV Status

The *rs7521189* genotype frequencies in the normotensive group were GG- 55 (27.36%), GA- 89 (44.28%), and AA- 57 (28.36%) compared to GG- 30 (29.41%), GA- 52 (50.98%), and AA- 20 (19.61%) in the EOPE group, irrespective of HIV status. The allele frequencies G and A were 199 (49.50%) and 203 (50.50%) in the normotensive group compared to 112 (54.90%) and 92 (45.10%) in the EOPE group, irrespective of HIV status (Appendix A).

The genotypic frequency association of co-dominant GG vs. AA, GG vs. GA, and GA vs. AA showed no significant association between the normotensive and the EOPE group, irrespective of HIV status. Additionally, comparisons of the dominant (GG vs. GA + AA), recessive (GG + GA vs. AA), and over-dominant (GG + AA vs. GA) alleles showed no significant association between the EOPE compared to the LOPE group (Table 3).

#### 2.4.6. Normotensive vs. Late Onset Preeclampsia Group Irrespective of HIV Status

The *rs7521189* genotype frequencies in the normotensive group were as follows: GG- 55 (27.36%), GA- 89 (44.28%), and AA- 57 (28.36%). The LOPE group exhibited frequencies of GG- 37 (36.27%), GA- 47 (46.08%), and AA- 18 (17.65%), regardless of HIV status. The allele frequencies for G and A were 199 (49.50%) and 203 (50.50%) in the normotensive group, while in the LOPE group, they were 121 (59.31%) and 83 (40.69%), regardless of HIV status (Appendix A).

The genotypic frequency association of the co-dominant gene polymorphism GG compared to AA demonstrated a significant correlation between the normotensive and LOPE groups, regardless of HIV status [OR = 2.130; 95% CI (1.085–4.181); adjusted *p* = 0.0317]. The genotypic frequencies of GG versus GA and GA versus AA exhibited no significant association between the normotensive and LOPE groups, regardless of HIV status. The genotypic frequency of the dominant alleles GG compared to GA + AA [OR = 1.847; 95% CI (1.019–3.347); adjusted *p* = 0.0485] indicated a significant association between the normotensive and LOPE groups, independent of HIV status. Furthermore, comparisons of the recessive (GG + GA vs. AA) and over-dominant (GG + AA vs. GA) alleles revealed no significant association between the normotensive group and the LOPE group. The allelic frequency association of G and A demonstrated a significant difference between normotensive individuals and the LOPE group, regardless of HIV status [OR = 1.487; 95% CI (1.057–2.092); adjusted *p* = 0.0252] (Table 3).

#### 2.4.7. HIV-Negative Group vs. HIV-Positive Group Regardless of Pregnancy Type

The *rs7521189* genotype frequencies of GG- 58 (28.57%), GA- 93 (45.81%), and AA- 52 (25.62%) in HIV-negative compared to GG- 64 (31.68%), GA- 95 (47.03%), and AA- 43 (21.29%) in the HIV-positive group, irrespective of pregnancy type. The allelic frequencies G and A were 209 (51.48%) and 197 (48.52%) in the HIV-negative group compared to 223 (55.20%) and 181 (44.80%) in the HIV-positive group, irrespective of pregnancy type (Appendix A).

The genotypic frequency association of gene polymorphism co-dominant GG vs. AA, GG vs. GA, and GA vs. AA exhibited no significant association between HIV-negative compared to HIV-positive groups. The dominant (GG vs. GA + AA), recessive (GG + GA vs. AA), and over-dominant (GG + AA vs. GA) alleles also showed no significant association between HIV-negative compared and HIV-positive pregnant women. Also, the allelic frequency association of G vs. A showed no significant difference between HIV-negative and HIV-positive groups (Table 3).

### 2.5. rs805305

#### 2.5.1. Normotensive vs. Preeclamptic HIV-Negative Group

The genotype frequencies of *rs805305* were CC- 73 (71.57%), CG- 23 (22.55%), and GG- 6 (5.88%) in the normotensive HIV-negative group, while in the preeclamptic HIV-negative group, the frequencies were CC- 66 (65.35%), CG- 22 (21.78%), and GG- 13 (12.87%). The allelic frequencies of C and G were 169 (82.84%) and 35 (17.16%) in the normotensive HIV-negative group, respectively, compared to 154 (76.24%) and 48 (23.76%) in the preeclamptic HIV-negative group. A substantial disparity existed among the genotypes for *rs805305* (*p* < 0.0001) (Appendix A; Figure 1). 

The findings indicate no significant association among the genotypic frequencies of CC vs. GG, CC vs. CG, and CG vs. GG in both the normotensive HIV-negative group and the preeclamptic HIV-negative group. Furthermore, analyses of the dominant (CC vs. CG + GG), recessive (CC + CG vs. GG), and over-dominant (CC + GG vs. CG) alleles revealed no significant associations between the normotensive HIV-negative and preeclamptic HIV-negative groups. The allelic frequency association between C and G did not exhibit a significant difference between the two groups (Table 4).

#### 2.5.2. Normotensive vs. Preeclamptic HIV-Positive

The genotype frequencies of *rs805305* were CC- 57 (57.58%), CG- 33 (33.33%), and GG- 9 (9.09%) in normotensive HIV-positive pregnant women compared to CC- 65 (63.11%), CG- 28 (27.18%), and GG- 10 (9.71%) in preeclamptic HIV-positive women. The allele frequencies C and G were 147 (74.24%) and 51 (25.76%) in the normotensive HIV-positive group and 158 (76.70%) and 48 (23.30%) in the preeclamptic HIV-positive group (Appendix A). 

The associations of genotypic and allelic frequencies of gene polymorphisms in normotensive women who are HIV-positive compared to the preeclamptic HIV-positive group showed no significant associations between CC vs. GG, CC vs. CG, and CG vs. GG. Similarly, the dominant (CC vs. CG + GG), recessive (CC + CG vs. GG) and over-dominant (CC + GG vs. CG) comparisons showed no significant associations between the normotensive HIV-positive compared to the preeclamptic HIV-positive group (Table 4).

#### 2.5.3. Normotensive Group vs. Preeclamptic Group Regardless of HIV Status

The frequencies of the *rs805305* genotype in the normotensive group were CC- 130 (64.68%), CG- 56 (27.86%), and GG- 15 (7.46%). In comparison, the preeclamptic group exhibited frequencies of CC- 131 (64.22%), CG- 50 (24.51%), and GG- 23 (11.27%), irrespective of HIV status. The allele frequencies for C and G were observed at 316 (78.61%) and 86 (21.39%) in the normotensive group, while in the preeclamptic group, they were 312 (76.47%) and 96 (23.53%), regardless of HIV status (Appendix A). 

The genotypic frequency association of gene polymorphism co-dominant CC vs. GG, CC vs. CG, and CG vs. GG demonstrated no significant association between normotensive pregnant women and the preeclamptic group, regardless of HIV status. Additionally, comparisons of the dominant (CC vs. CG + GG), recessive (CC + CG vs. GG), and over-dominant (CC + GG vs. CG) alleles revealed no significant association between the normotensive and preeclamptic groups, regardless of HIV status. The allelic frequency association of C and G did not demonstrate a significant difference between normotensive pregnant women and the preeclamptic group, regardless of HIV status (Table 4).

#### 2.5.4. Early vs. Late Onset Preeclampsia Groups Regardless of HIV Status

The *rs805305* genotype frequencies in the EOPE group were CC- 67 (65.69%), CG- 21 (20.59%), and GG- 14 (13.73%) compared to CC- 64 (62.75%), CG- 29 (28.43%), and GG- 9 (8.82%) in the LOPE group, irrespective of HIV status. The allele frequencies C and G were 155 (75.98%) and 49 (24.02%) in the EOPE group compared to 157 (76.96%) and 47 (23.04%) in the LOPE group, irrespective of HIV status (Appendix A). 

The analysis of genotypic frequency associations for gene polymorphism, comparing co-dominant CC vs. GG, CC vs. CG, and CG vs. GG, revealed no significant association between the EOPE and LOPE groups, regardless of HIV status. In addition, comparisons of the dominant (CC vs. CG + GG), recessive (CC + CG vs. GG), and over-dominant (CC + GG vs. CG) alleles revealed no significant association between the EOPE and LOPE groups. The allelic frequency association of C and G did not exhibit a significant difference in EOPE when compared to the LOPE group, regardless of HIV status (Table 4).

#### 2.5.5. Normotensive Group vs. Early Onset Preeclampsia Group Regardless of HIV Status

The *rs805305* genotype frequencies in the normotensive group were CC- 130 (64.68%), CG- 56 (27.86%), and GG- 15 (7.46%) compared to CC- 67 (65.69%), CG- 21 (20.59%) and GG- 14 (13.73%) in the EOPE group, regardless of HIV status. The allele frequencies C and G were 316 (78.61%) and 86 (21.39%) in the normotensive group compared to 155 (75.98%) and 49 (24.02%) in the EOPE group, regardless of HIV status (Appendix A). 

The genotypic frequency association of gene polymorphism co-dominant CC vs. GG, CC vs. CG, and CG vs. GG showed no significant association between the normotensive and the EOPE group irrespective of HIV status. Additionally, comparisons of the dominant (CC vs. CG + GG), recessive (CC + CG vs. GG), and over-dominant (CC + GG vs. CG) alleles showed no significant association between the normotensive group and the EOPE group. Also, the allelic frequency association of C and G showed no significant difference in the normotensive group compared to the EOPE group, irrespective of HIV status (Table 4).

#### 2.5.6. Normotensive vs. Late Onset Preeclampsia Group Regardless of HIV Status

The *rs805305* genotype frequencies in the normotensive group were CC- 130 (64.68%), CG- 56 (27.86%), and GG- 15 (7.46%), but in the LOPE group, they were CC- 64 (62.75%), CG- 29 (28.43%), and GG- 9 (8.82%), regardless of HIV status. The allele frequencies C and G were 316 (78.61%) and 86 (21.39%) in the normotensive group, compared to 157 (76.96%) and 47 (23.04%) in the LOPE group, regardless of HIV status (Appendix A). 

The genotypic frequency associations of gene polymorphism co-dominant CC vs. GG, CC versus CG, and CG versus GG exhibited no significant correlation between the normotensive and LOPE groups, regardless of HIV status. Furthermore, assessments of the dominant (CC vs. CG + GG), recessive (CC + CG vs. GG), and over-dominant (CC + GG vs. CG) alleles revealed no significant connection between the normotensive and LOPE groups. The allelic frequency association of C and G exhibited no significant difference between normotensive individuals and the LOPE group, regardless of HIV status (Table 4).

#### 2.5.7. HIV-Negative Group vs. HIV-Positive Group Regardless of Pregnancy Type

The *rs805305* genotype frequencies in the HIV-negative group were CC- 139 (68.47%), CG- 45 (22.17%), and GG- 19 (9.36%) compared to CC- 122 (60.40%), CG- 61 (30.20%) and GG- 19 (9.41%) in the HIV-positive group, irrespective of pregnancy type. The allelic frequencies C and G were 323 (79.56%) and 83 (20.44%) in an HIV-negative group compared to 305 (75.50%) and 99 (24.50%) in the HIV-positive group, regardless of pregnancy type (Appendix A). 

The genotypic frequency associations of gene polymorphism co-dominant CC vs. GG, CC vs. CG, and CG vs. GG demonstrated no significant correlation between HIV-negative pregnant women and the HIV-positive group, regardless of pregnancy type. Furthermore, comparisons of the dominant (CC vs. CG + GG), recessive (CC + CG vs. GG), and over-dominant (CC + GG vs. CG) alleles revealed no significant connection between the HIV-negative and HIV-positive groups. The allelic frequency association of C and G showed no significant difference between the HIV-negative and HIV-positive groups, irrespective of the pregnancy type (Table 4). 

### 2.6. rs3131383

#### 2.6.1. Normotensive HIV-Negative vs. Preeclamptic HIV-Negative Groups

The genotype frequencies of *rs3131383* were GG- 71 (69.61%), GT- 16 (16.69%), and TT- 15 (14.71%) in the normotensive HIV-negative group, while in the preeclamptic HIV-negative group, the frequencies were GG- 54 (53.47%), GT- 26 (25.74%), and TT- 21 (20.79%). The allele frequencies of G and T were 158 (77.45%) and 46 (22.55%) in the normotensive HIV-negative group, respectively, compared to 134 (66.34%) and 68 (33.66%) in the preeclamptic HIV-negative group. There was a significant difference between all three means of genotypes for *rs3131383* (*p* < 0.001) (Appendix A; Figure 1).

The genotypic frequency association of gene polymorphism in co-dominant alleles between normotensive HIV-negative individuals and preeclamptic HIV-negative individuals revealed no significant association between the genotypic frequencies of GG vs. TT and GT vs. TT. There was a significant association in the genotypic frequencies of GG vs. GT [OR = 2.137; 95% CI (1.044–4.374); adjusted *p* = 0.0490] between the normotensive HIV-negative group and the preeclamptic HIV-negative group. Additionally, comparisons between the dominant GG and the combined GT + TT alleles [OR = 1.993; 95% CI (1.121–3.544); adjusted *p* = 0.0212] indicated a significant association between the normotensive HIV-negative group and the preeclamptic HIV-negative group. The recessive (GG + GT vs. TT) and over-dominant (GG + TT vs. GT) alleles exhibited no significant associations between the normotensive HIV-negative group and the preeclamptic HIV-negative group. The allelic frequency association between G and T demonstrated a significant difference between the normotensive HIV-negative group and the preeclamptic HIV-negative group [OR = 1.743; 95% CI (1.123–2.705); adjusted *p* = 0.0150] (Table 5).

#### 2.6.2. Normotensive HIV-Positive Group vs. Preeclamptic HIV-Positive Group

The genotype frequencies of *rs3131383* were GG- 66 (66.67%), GT- 20 (20.20%), and TT- 13 (13.13%) in the normotensive HIV-positive group compared to GG- 65 (63.11%), GT- 22 (21.36%), and TT- 16 (15.53%) in the preeclamptic HIV-positive group. The allele frequencies G and T were 152 (76.77%) and 46 (23.23%) in normotensive HIV-positive women compared to 152 (73.79%) and 54 (26.21%) in the preeclamptic HIV-positive women group (Appendix A).

The genotypic frequency association of gene polymorphism in normotensive HIV-positive women compared to preeclamptic HIV-positive women shows no significant association between the genotypic frequencies of GG vs. TT, GG vs. GT, and GT vs. TT. Additionally, comparisons of the dominant (GG vs. GT + TT), recessive (GG + GT vs. TT), and over-dominant (GG + TT vs. GT) alleles revealed no significant associations between the normotensive HIV-positive group and the preeclamptic HIV-positive group. The allelic frequency association between G and T did not demonstrate a significant difference between the normotensive HIV-positive group and the preeclamptic HIV-positive group (Table 5).

#### 2.6.3. Normotensive vs. Preeclamptic Groups Irrespective of HIV Status

The *rs3131383* genotype frequencies in normotensive pregnant women were GG- 137 (68.16%), GT- 36 (17.91%), and TT- 28 (13.93%), but in preeclamptic women, the frequencies were GG- 119 (58.33%), GT- 48 (23.53%), and TT- 37 (18.14%), regardless of HIV status. The allele frequencies for G and T were 310 (77.11%) and 92 (22.89%) in the normotensive group, compared to 286 (70.10%) and 122 (29.90%) in the preeclamptic group, regardless of HIV status (Appendix A). 

The genotypic frequency associations of gene polymorphism co-dominant GG vs. TT, GG vs. GT, and GT vs. TT presented no significant correlation between the normotensive and preeclamptic groups, regardless of HIV status. Comparisons of the dominant (GG vs. GT + TT), recessive (GG + GT vs. TT), and over-dominant (GG + TT vs. GT) alleles indicated no significant association between the normotensive group and the preeclamptic group. The allelic frequency association of G and T demonstrated a significant difference between the normotensive and preeclamptic groups, regardless of HIV status [OR = 1.437; 95% CI (1.049–2.289); adjusted *p* = 0.0257] (Table 5).

#### 2.6.4. Early Onset Preeclampsia vs. Late Onset Preeclampsia Groups Regardless of HIV Status

The genotype frequencies of *rs3131383* in the EOPE group were GG- 68 (66.67%), GT- 21 (20.59%), and TT- 13 (12.75%). In contrast, the LOPE group exhibited frequencies of GG- 51 (50.00%), GT- 27 (26.47%), and TT- 24 (23.53%), regardless of HIV status. The allele frequencies of G and T were 157 (76.96%) and 47 (23.04%) in the EOPE group, compared to 129 (63.24%) and 75 (36.76%) in the LOPE group, regardless of HIV status (Appendix A). 

The genotypic frequency association of co-dominant GG compared to TT was significantly different in the EOPE group compared to the LOPE group, regardless of HIV status [OR = 2.462; 95% CI (1.144–5.298); adjusted *p* = 0.0239]. Nonetheless, the genotypic frequencies of GG compared to GT and GT compared to TT did not show a significant association between the EOPE and LOPE groups, regardless of HIV status. The dominant alleles GG vs. GT + TT [OR = 2.000; 95% CI (1.136–3.522); adjusted *p* = 0.0228] exhibited a significant association between the EOPE group and the LOPE group, regardless of HIV status. Furthermore, the recessive (GG + GT vs. TT) and over-dominant (GG + TT vs. GT) alleles exhibited no significant association between the EOPE and LOPE groups. The allelic frequency association of G and T demonstrated a significant difference between the EOPE and LOPE groups, regardless of HIV status [OR = 1.942; 95% CI (1.260–2.994); adjusted *p* = 0.0034] (Table 5).

#### 2.6.5. Normotensive Group vs. Early Onset Preeclampsia Group Regardless of HIV Status

The *rs3131383* genotype frequencies in the normotensive cohort were GG- 137 (68.16%), GT- 36 (17.91%), and TT- 28 (13.93%), but in the EOPE group, they were GG- 68 (66.67%), GT- 21 (20.59%), and TT- 13 (12.75%), regardless of HIV status. The allele frequencies for G and T were 310 (77.11%) and 92 (22.89%) in the normotensive group, respectively, compared to 157 (76.96%) and 47 (23.04%) in the EOPE group, regardless of HIV status (Appendix A). 

The genotypic frequency connections of co-dominant GG vs. TT, GG versus GT, and GT versus TT demonstrated no significant correlation between the normotensive and EOPE groups, irrespective of HIV status. Additionally, comparisons of the dominant (GG vs. GT + TT), recessive (GG + GT vs. TT), and over-dominant (GG + TT vs. GT) alleles showed no significant association between the EOPE compared to the LOPE group. Also, the allelic frequency association of G and T showed no significant difference in the EOPE vs. the LOPE group, irrespective of HIV status (Table 5).

#### 2.6.6. Normotensive Group vs. Late Onset Preeclampsia Group Regardless of HIV Status

The *rs3131383* genotype frequencies in the normotensive group were GG- 137 (68.16%), GT- 36 (17.91%), and TT- 28 (13.93%). In contrast, the LOPE group exhibited frequencies of GG- 51 (50.00%), GT- 27 (26.47%), and TT- 24 (23.53%), regardless of HIV status. The allele frequencies for G and T were 310 (77.11%) and 92 (22.89%) in the normotensive group, compared to 129 (63.24%) and 75 (36.76%) in the LOPE group, regardless of HIV status (Appendix A). 

The genotypic frequency association of gene polymorphism co-dominant GG vs. TT [OR = 2.303; 95% CI (1.223–4.337); adjusted *p* = 0.0113] and GG vs. GT [OR = 2.015; 95% CI (1.113–3.648); adjusted *p* = 0.0271] demonstrated a significant association between the normotensive and LOPE groups, regardless of HIV status. The genotypic frequency of GT compared to TT did not demonstrate a significant association between the normotensive and LOPE groups, regardless of HIV status. Additionally, comparisons of the dominant GG vs. GT + TT [OR = 2.141; 95% CI (1.313–3.490); adjusted *p* = 0.0026] allele indicated a significant association between the normotensive group and the LOPE group. The recessive (GG + GT vs. TT) and over-dominant (GG + TT vs. GT) alleles exhibited no significant association when comparing the normotensive group to the LOPE group. The allelic frequency association of G and T demonstrated a significant difference between normotensive pregnant women and the LOPE group, regardless of HIV status [OR = 1.449; 95% CI (0.9229–2.254); adjusted *p* = 0.0004] (Table 5).

#### 2.6.7. HIV-Negative Groups vs. HIV-Positive Groups Regardless of Pregnancy Type

The *rs3131383* genotype frequencies in the HIV-negative group were GG- 125 (61.58%), GT- 42 (20.69%), and TT- 36 (17.73%). In comparison, the HIV-positive group exhibited frequencies of GG- 131 (64.85%), GT- 42 (20.79%), and TT- 29 (14.36%), regardless of pregnancy type. In the HIV-negative group, the allele frequencies of G and T were 292 (71.92%) and 114 (28.08%), respectively, while in the HIV-positive group, they were 304 (75.25%) and 100 (24.75%), irrespective of pregnancy type (Appendix A). 

The genotypic frequency association of gene polymorphism co-dominant GG vs. TT, GG vs. GT, and GT vs. TT revealed no significant association between the HIV-negative and HIV-positive groups, regardless of pregnancy type. Furthermore, analyses of the dominant (GG vs. GT + TT), recessive (GG + GT vs. TT), and over-dominant (GG + TT vs. GT) alleles revealed no significant association between the HIV-negative and HIV-positive groups. The allelic frequency association of G and T did not demonstrate a significant difference between the HIV-negative and HIV-positive groups, regardless of the type of pregnancy (Table 5).

## 3. Discussion

This novel study reports that genetic variations of *rs669173*, *rs7521189*, and *rs3131383* of the *DDAH* gene were significantly associated with PE comorbid with HIV infection in women of African ancestry. Of note, *DDAH 1* and *2* encode the enzyme that controls the degradation of endogenous inhibitors of NOS-ADMA [31]. Consequently, genetic variation in the *DDAH* genes will affect downstream activity and synthesis of ADMA with resultant dysfunction of vascular NO response which results in the clinical manifestations of PE (Figure 2) [41,42,43]. 

### 3.1. DDAH 1

#### 3.1.1. ADMA *rs669173*

This study reports a significant association between the *rs669173* polymorphism, and the genotypic frequencies of TT compared to CC among *rs669173* variations (TT, TC, CC) in normotensive HIV-negative pregnant women compared to their preeclamptic HIV-negative counterparts. The association of allelic frequency between T and C demonstrated a significant correlation between normotensive pregnant HIV-negative women and preeclamptic HIV-negative women. The findings suggest that the *rs669173* T>C polymorphism is associated with an increased risk of developing PE in HIV-negative pregnant women of African descent. Similar to our study, research on Finnish women demonstrated a correlation between *DDAH 1* gene polymorphisms and endothelial dysfunction in the risk of PE development [34]. We also report no significant associations were found for the dominant, recessive, and/or over-dominant alleles of TT vs. TC + CC, TT + TC vs. CC, or TT + CC vs. TC. The T and C alleles were significantly different between normotensive pregnant and PE women, irrespective of HIV status.

The homozygous TT alleles were observed to be more prevalent in the normotensive group than in PE, independent of HIV status. Of note, our study found that preeclamptic women of African ancestry are more likely to have both a heterozygous TC and a homozygous CC genotype advantage, influencing the biological mechanism underlying and balancing natural selection [44]. In contrast to our findings, a study in a Caucasian population reported an uncommon mutation (c.260C>T) in the coding sequence of exon 1 of the *DDAH 1* gene. This mutation results in an exchange of amino acids (Thr87Met), and it increases the risk of hypertension and cardiovascular disease [45]. Studies performed on Australian, Egyptian, and Swedish populations have demonstrated that genetic variations of *DDAH 1* genes are related to ADMA levels [40,46,47,48]. Abhary et al. (2010) reported that the *DDAH 1 rs669173* C allele was significantly associated with a lower plasma ADMA level in a large Australian cohort with type 2 diabetes [40]. Another study using 70-year-old Swedish participants who were part of a vasculature investigation also showed the *rs669173* C allele to be significantly associated with lower ADMA levels [46]. Interestingly, it may be proposed that individuals with the *rs669173* C allele exhibit heightened NOS activity [49], which is a variation that opposes the levels of ADMA since there is a negative correlation between vascular NO bioavailability and serum ADMA levels [29]. Therefore, one may deduce that *rs669173* T>C polymorphism may have an impact on the development of PE in women of African descent. 

In our study, the dominant allele, in contrast to the recessive and over-dominant allele frequency associations, exhibited a significant association between the normotensive HIV-negative and preeclamptic HIV-negative groups. ADMA can reduce NO in vitro and in vivo by competitively inhibiting three isoforms of NOS activity [27,29,50]. NOS is responsible for the catalysis of NO, which plays a crucial role in maintaining vascular structure and function [51,52]. To our knowledge, this is the first study to report *rs669173* polymorphisms in PE. However, our findings also show that the presence of HIV infection does not affect the genetic variation of *rs669173* of the *DDAH 1* gene in women with PE.

Animal studies have provided evidence for the metabolic regulation of ADMA by *DDAH* genes and their impact on the integrity of endothelial cells [32,53,54,55,56]. A study of transgenic mice overexpressing *DDAH 1* showed that a two-fold decrease in plasma ADMA correlated with a two-fold increase in tissue NOS activity [56]. On the other hand, Leiper et al. (2007) and Wojciak-Stothard et al. (2009) reported that in *DDAH 1* gene knockout mice, there was an accumulation of ADMA and inhibition of NO pathways due to the DDAH impediment leading to endothelial impairment [57,58]. Similarly, our study found that *rs669173* T>C polymorphism may be linked with an increase in ADMA levels which inhibits NOS, thus pre-empting endothelial impairment and predisposing PE in our black population (Figure 2) [59]. These findings corroborate several other studies that found elevation of serum ADMA associated with a decrease in NO production in vascular endothelial cells leading to the development of diabetic macrovascular and microvascular problems, including retinopathy, nephropathy, and atherosclerosis [60,61,62,63]. Another study by Hanneman et al. (2020) reported that chronic hypoxia causes an increase in ADMA levels which then affects NO production and leads to endothelial impairment and vasoconstriction [64]. Of note, genetic variation of *DDAH 1* occurs in intron 1 of the gene [40]. Our findings suggest that there was no link to *rs669173* T>C polymorphism in pregnant women with HIV infection. Notably, ART was administered to every HIV-infected woman in our study; however, the association between *rs669173* T>C and ART responses or outcomes has not been thoroughly studied. 

#### 3.1.2. ADMA *rs7521189*

The allelic frequency association between G and A alleles showed no significant disparity between normotensive and PE, regardless of HIV status. Moreover, there was a significant association between normotensive and preeclamptic groups, irrespective of HIV status for the co-dominant GG vs. AA, while GG vs. GA and GA vs. AA were not significant. This result indicates that there may be a significant association between the *rs7521189* G>A polymorphisms and PE development in women of African ancestry. A previous study reported the *rs7521189* polymorphism association with the serum ADMA level in patients with type 2 diabetes without retinopathy [40]. However, we did not find any study that demonstrated this polymorphism that directly demonstrated an association with HIV infection comorbid with PE. Additionally, the allelic frequency association between G and A and the dominant allele GG vs. GA + AA were also significantly different, showing a strong correlation between *rs7521189* and PE in the studied population. Interestingly, this is the first study to evaluate *rs7521189* polymorphisms in women with PE and HIV infection in an African population.

The genotype frequency analysis revealed a significant association between GG and AA in the normotensive group compared to the LOPE group, irrespective of HIV status, while no significant associations were observed for the subsequent co-dominant alleles. The dominant allele GG exhibited a significant difference compared to GA + AA, as well as a notable distinction in allelic frequency between G and A when comparing normotensive pregnant women to the LOPE group. This significant difference could indicate a possible link between the *rs7521189* variant and the predisposition to LOPE in women of African ancestry. Neutral mutations may cause most late-onset diseases and can completely replace ancestral alleles. Consequently, the ancestral (rare) allele serves as the protective allele, whereas the derived (common) allele functions as the risk allele [65]. Furthermore, this study revealed that individuals with PE were more likely to have GA alleles, while AA alleles were more common in normotensive individuals, regardless of HIV status. At present, there are no studies examining the association between the heterozygous genotype and the development of PE. However, our research suggests that individuals with the GA genotype may influence the risk of PE development in women of African ancestry. A previous study examining both men and women (50% per group) from a Swedish population reported that *rs7521189* polymorphism correlates with circulating levels of ADMA but not the measurement of endothelial-dependent vasodilation or flow-mediated dilation [46]. Our study shows no significant genotypic frequency association between HIV-negative and HIV-positive pregnant women for GG vs. AA and GA vs. AA. Also, the allelic frequency association (G vs. A) showed no significant difference between the two groups. The findings indicate that the *rs7521189* polymorphism does not exert a genetic influence on HIV susceptibility in pregnant women.

### 3.2. DDAH 2

#### 3.2.1. ADMA *rs805305*

We report no significant relationships in genotypic frequencies between normotensive HIV-negative and preeclamptic HIV-negative individuals. Likewise, the allelic frequency connection between the C and G alleles exhibited no significant difference between the normotensive and preeclamptic groups. The findings suggest that the *rs805305* polymorphism did not influence the development of PE. However, previous studies have indicated that the carriers of the major allele of *DDAH 2* (*rs805304*) promote an increase in ADMA during hypoxia, thus influencing PE development [30].

Analysis of genotype frequencies in normotensive HIV-positive individuals against preeclamptic HIV-positive individuals revealed no significant relationships when comparing CC vs. GG, CC vs. CG, and CG vs. GG in both groups. The allelic frequency correlation between C and G exhibited no significant difference. Similarly, a study using a Caucasian population reported no correlation between -449G polymorphism and ADMA levels; nonetheless, there was an association with an increased prevalence of hypertension [66]. Subsequent research employing dominant, recessive, and over-dominant allele models likewise failed to demonstrate significant relationships between the normotensive and preeclamptic groups. Our data indicate that the *rs805305* polymorphism is not linked to an elevated risk of preeclampsia in HIV-positive pregnant women of African descent. This research is corroborated by Khaliq et al. (2020) and Mlambo et al. (2024), who also showed that homozygous (CC and GG) genotypes were more prevalent among preeclamptic women of African descent, indicating a heterozygous disadvantage regardless of HIV status; normotensive groups had higher levels of the heterozygous genotype and GC alleles than preeclamptic groups [67,68]. In contrast to our findings, Yusuf et al. (2013) investigated the *DDAH 2* SNP *rs805305* (–449 G/C; promoter region) in 35 hemodialysis patients with hypertension, reporting that the GG homozygote had the highest ADMA levels in an Indonesian population [69]. These findings were also corroborated in a study by Mendes et al. (2024), who observed elevated ADMA levels in Brazilian women with EOPE that carried the GC and GG genotypes for the *DDAH 2* -449G/C *rs805305* polymorphism [70]. This was found to be a common genetic variant in the *DDAH 2* promoter region that affected the production of *DDAH 2* within endothelial cells [71]. Although the impact of *DDAH 2* activity on ADMA levels in pregnant African women has not been investigated, we believe that the *rs805305* polymorphism does not directly alter *DDAH 2* activity and hence may not be directly linked to the development of PE. A study in an Irish population reported that the presence of a G at position -449 in the promoter region of the *DDAH 2* gene correlates with elevated ADMA levels, indicating that the *DDAH 2* gene with a G compared to a C at this locus had reduced activity [72,73]. Also, high ADMA levels are a result of *DDAH 2* activity deficiency, thus inhibiting NO production [74]. Hence, such maladaptation in NO production and NO signaling plays a vital role in the progression and severity of PE (Figure 2) [75,76,77]. 

Additional evidence indicates that −449 G/C *DDAH 2* variants may potentially inhibit the development of intracerebral hemorrhage [78]. Another study in an American population identified a genetic polymorphism, specifically the presence of at least one G allele at the -449 position within the *DDAH 2* gene, which correlates with low plasma ADMA concentrations in children experiencing severe sepsis or septic shock, in contrast to non-septic controls [79]. Furthermore, a study by Azevedo et al. (2017) also reported that *rs805305* polymorphisms may be markers for sildenafil citrate responsiveness in post-prostatectomy erectile dysfunction [80]. Our novel findings indicate that *rs805305* polymorphism is not associated with PE development in women of African ancestry. Furthermore, these results highlight that *rs805305* polymorphism has no genetic impact on HIV susceptibility during pregnancy.

#### 3.2.2. ADMA *rs3131383*

We observed notable associations in the genotypic frequencies of GG compared to GT, while no associations were found between GG and TT, as well as GT and TT. Comparisons of dominant alleles reveal notable associations between normotensive and preeclamptic HIV-negative groups, whereas the frequencies of recessive and co-dominant alleles did not show significant differences. The allelic frequency association between G and T demonstrated a significance between normotensive pregnant HIV-negative and preeclamptic HIV-negative women. These findings indicate a significant association between the *rs3131383* polymorphism and an increased risk of PE development, regardless of HIV status in women of African descent. A study reported a strong correlation between SNPs of *rs3131383* of the *DDAH 2* gene with ADMA concentration in patients from a Caucasian cohort with type 2 diabetes [40]. 

The G and T alleles showed a significant correlation between the normotensive and preeclamptic groups, irrespective of HIV status. These findings imply that a higher risk of PE propensity may be linked to the G allele. According to our knowledge, the *rs3131383* polymorphism has not been studied in PE. However, a study in the Tunisian population identified the *DDAH* 2 G allele as a potential genetic predictor for coronary artery disease development [81]. A characteristic feature of coronary artery disease is an alteration in NO metabolism, a major factor in the etiology of PE [82,83,84]. The endothelium produces NO, a significant vasodilator factor with anti-thrombotic, anti-inflammatory, and anti-proliferative properties [85]. The ensuing decrease in NO production in vascular endothelial cells contributes to diabetic vascular complications, including retinopathy, nephropathy, PE, and atherosclerosis. Elevation of ADMA has also been linked to several other diseases [43,61,63,86]. In the normotensive compared to LOPE groups, we demonstrated significant associations in GG vs. TT and GG vs. GT genotypic frequencies. The allelic frequency association between G and T and the dominant alleles in the normotensive compared to LOPE groups were also significantly different. The findings from our study denote a significant relationship between the *rs3131383* polymorphism and an increased risk of LOPE in pregnant women of African descent. 

This study reports differences in the *rs3131383* genotype of the *DDAH 2* gene, along with variations in allele frequency between EOPE and LOPE women, regardless of their HIV status. Overexpression of *DDAH 2* triggers mRNA expression and secretion of VEGF, thereby increasing proliferation and migration of endothelial cells [87]. Studies by Kupferminc et al. (1997) and Lee et al. (2007) reported that elevated maternal VEGF levels were observed in patients with PE, correlating with the severity of hypertension, indicating a potential role for VEGF in the pathogenesis of PE [88,89]. Similarly, our study reports that the *rs3131383* polymorphism affects DDAH 2 activity, leading to the accumulation of ADMA and endothelial impairment, thereby predisposing PE development (Figure 2). Regardless of HIV status, our study found differences in the frequency of the G and T alleles, as well as the GG, GT, and TT genotypes, between the EOPE and LOPE groups. Interestingly, women with EOPE had a higher incidence of the GG genotype compared to those with LOPE, indicating that those who carry this variant may be more likely to develop EOPE. In addition, there was a difference in the frequency of the G and T alleles between the two groups, with the G allele being more common. However, we know that the *rs3131383* polymorphism has not been studied in pregnant women. The complicated pathophysiology and type of PE, as well as the possible impact of genetic variation particular to a population and environmental circumstances, are all reflected in the differential allelic expression. Our results provide credence to the possible involvement of *rs3131383* polymorphism in developing PE.

Genetic variants that impair NO have been shown to increase hypertension risk [90]. In a systematic review of the association of 46 polymorphisms in 33 genes with hypertension, only *rs4340*, *rs699*, and *rs5186* demonstrated a link in African populations [91]. In Africa, gene-based analyses of 78 variants have shown an association of a component of the renin-angiotensin-aldosterone system with essential hypertension [92]. Moreover, genetic variations of *DDAH 1* genes were shown in an Egyptian group but not in pregnancy-related complications [48]. Thirty-two single-nucleotide polymorphisms of *DDAH 1* and *2* were examined in American patients with pulmonary hypertension in patients with bronchopulmonary dysplasia compared to bronchopulmonary dysplasia alone [93]. The authors report that only *rs480414* of the *DDAH 1* gene was protective against the development of pulmonary hypertension in patients with bronchopulmonary dysplasia [93]. The data on *DDAH* polymorphisms in African populations is incomplete, yet it parallels the relatively modest effects of NOS polymorphisms [94]. ]. It is possible that the previously reported *DDAH* SNPs showing association with ADMA levels in PE are not the variants directly involved in regulating ADMA but rather are in linkage disequilibrium with such functional SNPs, as our tag SNP approach may identify genetic associations but not necessarily the functional genetic variant. Furthermore, it is plausible that untyped SNPs in the current study affect HIV-related ADMA levels since HIV and PE are associated with inflammation and endothelial impairment [95,96,97]; however, there was no discernible correlation between HIV infection and any of the study groups. The findings indicate an absence of genetic influence from the *rs669173*, *rs7521189*, *rs805305*, and *rs3131383* polymorphisms on HIV susceptibility during pregnancy. Alternatively, it emphasizes that HIV-1 dependence genes identified using genome-wide short interfering RNA assays are notably conserved [98]. A limitation of our study is that we did not consider socioeconomic and environmental factors as confounders in ADMA polymorphisms in women of African ancestry. Also, another limitation of our study was the sample size (*n* = 405). There is a necessity for large-scale analysis across different ethnic groups to accurately characterize these SNPs in PE stratified by gestational age and comorbid with HIV infection [99].

We also report that PE women delivered at a lower gestational age than normotensive women. The severity of the problem is shown by the markedly raised diastolic and systolic blood pressure values in the PE group. Furthermore, a noteworthy distinction in maternal weight was observed between the normotensive and PE groups, bearing significance for better management. Future studies should incorporate routine genetic testing in all HIV patients, as this may be valuable for identifying individuals with specific SNPs that could influence their response to antiretroviral therapy.

## 4. Materials and Methods

### 4.1. Study Population and Design

Institutional ethical approval was obtained for using the archived samples (BCA338/17) in this prospective study (BE3255/21). All participants gave informed consent. Blood was obtained from pregnant women attending a large district hospital in eThekwini, KwaZulu-Natal, South Africa. Study groups included N pregnant women and PE women stratified by HIV status. PE was defined as sustained systolic blood pressure ≥ 140 mmHg and diastolic blood pressure 90 mmHg or greater, taken at least four hours apart after 20 weeks’ gestation. Proteinuria was defined as a urine protein concentration of ≥300 mg/dL or 1+ on a urine dipstick in at least two random specimens collected at least 4 h apart [1].

Inclusion criteria—The study group consisted of primigravid and multigravid individuals with PE (≥140/90 mmHg and the presence of a single occurrence of proteinuria). The control group consisted of pregnant women with normal blood pressure. A quick test was used to determine the mother’s HIV status. Regardless of their CD4 cell level, all HIV-positive trial participants received ART and prevention of mother-to-child transmission (PMTCT) treatment during pregnancy and breastfeeding. Women who had CD4 counts below 500 cells/mm3 were also treated with antiretroviral medication continuously after nursing. Women receiving antiretroviral treatment are given either a single medication, such as zidovudine, commonly known as azido-thymidine (AZT), or a combination of many medications, such as efavirenz (EFV), tenofovir disoprovil fumarate (TDF, Viread), and emtricitabine (FTC, Emtriva). In accordance with South African national HIV guidelines, several patients received PMTCT (nevirapine) along with an additional medication combination consisting of Abacavir (ABC, Ziagen), Lamivudine (3TC, Epivir), and Efavirenz (EFV). Nevirapine prophylaxis was administered to infants exposed to HIV for four to six weeks.

Exclusion criteria—All groups encompassed chorioamnionitis, chronic hypertension, eclampsia, and placental abruption; intrauterine death, pre-gestational diabetes, gestational diabetes, and chronic renal disease; systemic lupus erythematosus, sickle cell disease, and antiphospholipid antibody syndrome; as well as thyroid disease, cardiac disease, active asthma necessitating medication during pregnancy, and pre-existing seizure disorders.

### 4.2. Sample Size

A sample size of 405 participants was required. Two groups (PE vs. N or HIV^+^ vs. HIV^−^) were required to detect the moderate effect size of 0.4. All calculations are with 80% power and 95% probability achieved using G*Power statistical analysis [100]. The study population comprised women with PE (*n* = 204) and normotensive pregnant women (*n* = 201). The normotensive cohort was categorized by HIV status into N^−^ (*n* = 102) and N^+^ (*n* = 99). The PE group was evenly split into EOPE (*n* = 102) and LOPE (*n* = 102) and further stratified by HIV status into EOPE^−^ (*n* = 50), EOPE^+^ (*n* = 52), LOPE^−^ (*n* = 51), and LOPE^+^ (*n* = 51).

### 4.3. DNA Isolation

Peripheral venous blood was extracted and stored in EDTA anticoagulant tubes. DNA was isolated from 200 µL of whole blood using the QIAmp DNA Mini Kit (QIAGEN Sciences, Germantown, MD, USA) and eluted into 50 μL of Buffer AE (QIAGEN) as per the manufacturer’s blood spin protocol. The concentration and purity of DNA were assessed using a NanoDrop spectrophotometer by measuring absorbance at 260 nm (ThermoFisher Scientific, Waltham, MA, USA). Pure DNA will have an A260/280 absorbance ratio of 1.8. To prevent DNA degradation, extractions were performed on ice. After extraction, DNA was eluted with nuclease-free water and stored at −80 °C until genotyping analysis.

### 4.4. TaqMan Genotyping of ADMA Gene Polymorphisms

Four SNPs (*rs669173*, *rs7521189*, *rs805305*, and *rs3131383*) were amplified to detect specific polymorphisms from purified DNA samples using TaqMan genotyping master mix (Applied Biosystems by ThermoFisher Scientific, Foster City, CA, USA), according to the manufacturer’s protocol. Genotyping of SNPs was performed using the QuantStudio 7 real-time Flex PCR (Life Technologies, Carlsbad, CA, USA). The final reaction master mix comprised 0.25 µL of a 20X working stock of TaqMan SNP genotyping assay, 2.5 µL of 2X TaqMan universal master mix, and 3 µL of DNA, resulting in a total volume of 5.75 µL per well, which was dispensed into each well. Two fluorescently labeled primers were used in the TaqMan genotyping experiment to distinguish between two alleles of a particular SNP. One primer had a minor groove binder (MGB) and a non-fluorescent quencher on the 3′ ends, while the other primer had 6-carboxyfluorescein (6-FAM™) dye (a blue fluorophore) for the mutant allele and VIC® dye (a green fluorophore) for the wild-type allele on the 5′ ends. 

The PCR protocol was as follows: pre-read at 60 °C for 30 s, hold (initiation) at 95 °C for 10 min, followed by 40 cycles consisting of denaturation at 95 °C for 15 s and annealing at 60 °C for 1 min. The final extension was performed at 60 °C for 30 s. Allelic discrimination findings were examined using QuantStudio™ design data analysis software version 1.5.2 after PCR amplification using the QuantStudio 5 Real-time PCR equipment (Applied Biosystems by ThermoFisher Scientific). 

### 4.5. Genetic Modelling

Table 6 delineates the four genetic models employed to analyze the allelic and genotypic frequencies of the gene polymorphisms *rs669173*, *rs7521189*, *rs805305*, and *rs3131383* of the ADMA gene in the context of PE comorbidity with HIV infection. An allele is classified as dominant when one copy suffices for complete phenotypic expression. Co-dominance occurs when the effects of both alleles are equally observable. An allele is termed recessive when a single copy does not produce a noticeable phenotypic effect. A dominant allele in one genotype may be recessive when paired with another allelic variant, contingent upon the partner allele in the heterozygous genotype. The focal character influences the dominance relationship among alleles [101]. A pleiotropic allele may exhibit a recessive effect for one trait and a dominant effect for another. 

### 4.6. Statistical Analysis

For table demographics analysis, a one-way ANOVA test was performed using the GraphPad Prism 5 software (GraphPad Software, San Diego, CA, USA). Due to the non-parametric distribution of the women’s data found by normality testing, a Kruskal–Wallis test and Dunn’s multiple comparison post hoc test were conducted. The results were expressed as the median and interquartile range. A *p*-value < 0.05 was considered statistically significant. The Hardy–Weinberg equilibrium test was used to check for conformance to the observed frequencies of the genotypes. Frequency and percentage were used to describe the presence of the genotypes. Logistic regression analysis was used for the analysis of the association between the ADMA genotype and risk of severe PE. Subgroups were compared using the chi-squared test or Fisher’s exact test as appropriate. The strength of association was reported as odds ratios and 95% confidence intervals for categorical data and Wilcoxon rank-sum tests for numeric data. The Bonferroni correction test was conducted for multiple comparisons.

## 5. Conclusions

Our study shows that the *DDAH 1* gene (*rs669173*, *rs7521189*) and *DDAH 2* gene (*rs3131383*) SNPs correlate significantly with the risk of PE development in pregnant women of African descent. In contrast, there was no significant correlation between SNPs of *rs805305* of the *DDAH 2* gene in the development of PE. This genetic variation will therefore impact downstream function, resulting in elevated ADMA levels which will inhibit NOS activity, thus impairing endothelial function and provoking the clinical symptoms of PE. Also, we report no significant associations between HIV infection and both the *DDAH 1* and *DDAH 2* gene polymorphisms in women of African ancestry, as the HIV-1 dependency genes identified through genome-wide short interfering RNA (siRNA) screens are largely conserved. Nonetheless, the complex interplay of the *DDAH 1* and *DDAH 2* variants and the downstream effect on endothelial impairment in PE and HIV infection requires further investigation in other ethnic groups.

## Figures and Tables

**Figure 1 ijms-26-03271-f001:**
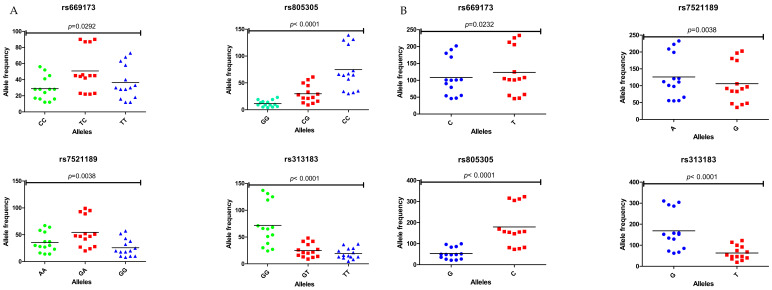
(**A**,**B**): Scatter plots depicting the distribution of genotypic and allelic frequencies for the SNPs *rs669173* (T/C), *rs7521189* (G/A), *rs805305* (C/G), and *rs3131383* (G/T) across the examined population. The x-axis represents alleles, while the y-axis indicates allelic frequency. Each point corresponds to a specific category or condition, with clustering patterns highlighting the relationship between genotype and allele frequencies. In (**A**), homozygous genotypes are represented in green and blue, while heterozygous genotypes are shown in red. Statistical significance was defined as *p* < 0.05.

**Figure 2 ijms-26-03271-f002:**
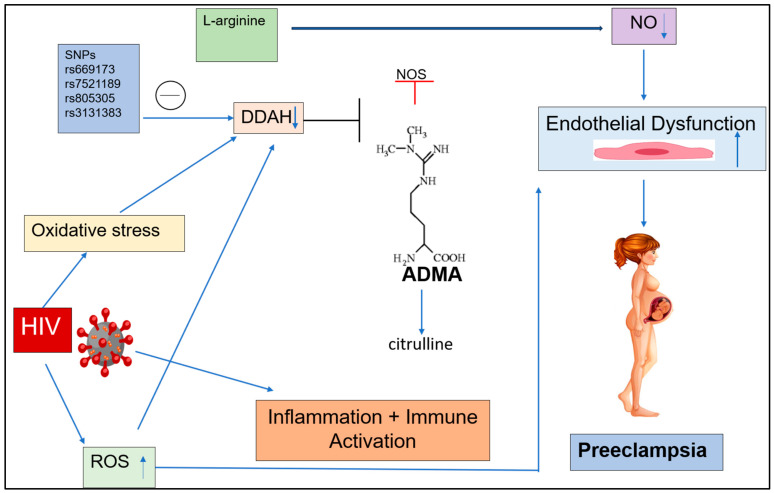
Schematic overview of the downstream effect of ADMA concentration in both preeclampsia and HIV. SNPs, oxidative stress, and ROS reduce DDAH activity, increasing ADMA levels. Elevated ADMA lowers NO bioavailability by competitively inhibiting NOS, leading to endothelial dysfunction and contributing to preeclampsia. DDAH, dimethylarginine dimethylaminohydrolase; ADMA, asymmetric dimethylarginine; SNP, single-nucleotide polymorphism; NO, nitric oxide; NOS, nitric oxide synthase; ROS, reactive oxygen species; HIV, human immunodeficiency virus.

**Table 1 ijms-26-03271-t001:** Demographic and clinical data of the study population (n = 405).

Patient Data	*p* Value	N^−^ (n = 102)	N^+^ (n = 99)	EOPE^−^ (n = 50)	EOPE^+^ (n = 52)	LOPE^−^ (n = 51)	LOPE^+^ (n = 51)
Systolic BP (mmHg) N vs. EOPE N vs. LOPE EOPE vs. LOPE	<0.0001 **** <0.0001 **** 0.9162	119.0 (111.0–124.0)	114.0 (109.0–120.0)	161.0 (155.0–168.0)	161.0 (154.0–165.0)	159.0 (155.0–168.8)	155.0 (148.0–164.0)
Diastolic BP (mmHg) N vs. EOPE N vs. LOPE EOPE vs. LOPE	<0.0001 **** <0.0001 **** 0.9543	71.00 (66.00–78.00)	71.00 (65.00–75.00)	104.0 (96.75–107.0)	104.0 (94.00–111.0)	101.5 (94.00–107.0)	99.00 (96.00–105.0)
Gestational Age (weeks) N vs. EOPE N vs. LOPE EOPE vs. LOPE	<0.0001 **** <0.0001 **** <0.0001 ****	39.00 (38.00–40.00)	38.00 (37.00–39.00)	30.00 (27.00–32.00)	29.00 (25.00–32.00)	38.00 (36.00–39.00)	37.00 (35.00–38.00)
Maternal Age (years) N vs. EOPE N vs. LOPE EOPE vs. LOPE	<0.0001 **** 0.9639 0.0068 **	23.00 (20.00–28.00)	27.00 (23.00–32.00)	30.00 (23.25–35.00)	30.00 (27.00–33.00)	24.00 (20.75–30.00)	29.00 (24.00–32.00)
Maternal Weight (Kg) N vs. EOPE N vs. LOPE EOPE vs. LOPE	0.1317 0.0012 ** 0.5743	71.50 (60.15–83.63)	70.00 (62.80–80.00)	73.00 (63.93–90.00)	73.10 (65.00–89.70)	73.00 (63.50–89.40)	77.00 (68.00–101.0)

N^−^, normotensive HIV-negative; N^+^, normotensive HIV-positive; EOPE^−^, early onset preeclampsia HIV-negative; EOPE^+^, early onset preeclampsia HIV-positive; LOPE^−^, late onset preeclampsia HIV-negative; LOPE^+^, late onset preeclampsia HIV-positive. All values are represented as median (IQR). Asterisks (*) denote significance: ** *p* < 0.01 and **** *p* < 0.0001.

**Table 2 ijms-26-03271-t002:** Genotypic and allelic associations of *ADMA* gene polymorphisms *rs669173* across pregnancy types stratified by HIV status.

SNP		N^−^ vs. PE^−^ OR (95% CI), *p* Value	N^+^ vs. PE^+^ OR (95% CI), *p* Value	EOPE^−^ vs. EOPE^+^ OR (95% CI), *p* Value	LOPE^−^ vs. LOPE^+^ OR (95% CI), *p* Value	N vs. PE OR (95% CI), *p* Value	HIV^−^ vs. HIV^+^ OR (95% CI), *p* Value	N vs. EOPE OR (95% CI), *p* Value	N vs. LOPE OR (95% CI), *p* Value	EOPE vs. LOPE OR (95% CI), *p* Value
*rs669173* T>C Genotype	TT vs. CC (Co-dominant)	0.4250 (0.1963–0.9202) *p* = 0.0352 *	0.7792 (0.3732–1.627) *p* = 0.5756	2.000 (0.7026–5.693) *p* = 0.2930	0.5625 (0.1951–1.622) *p* = 0.4230	0.5817 (0.3423–0.9886) *p* = 0.0605	1.247 (0.7370–2.111) *p* = 0.4246	0.6018 (0.3168–1.143) *p* = 0.1397	0.5616 (0.2936–1.075) *p* = 0.0971	0.9333 (0.4476–1.946) *p* ≥ 0.9999
	TT vs. TC (Co-dominant)	1.429 (0.7565–2.698) *p* = 0.3338	1.179 (0.6157–2.256) *p* = 0.7409	0.6667 (0.2605–1.706) *p* = 0.4788	1.333 (0.5178–3.433) *p* = 0.6343	1.302 (0.8272–2.049) *p* = 0.2993	0.9584 (0.6098–1.506) *p* = 0.9083	1.231 (0.7038–2.152) *p* = 0.4826	1.378 (0.7846–2.422) *p* = 0.3206	1.120 (0.5786–2.168) *p* = 0.8664
	TC vs. CC (Co-dominant)	0.6071 (0.2923 –1.261) *p* = 0.2030	0.9184 (0.4612–1.829) *p* = 0.8616	1.333 (0.5137–3.461) *p* = 0.6318	0.7500 (0.2913–1.931) *p* = 0.6343	0.7574 (0.4597–1.248) *p* = 0.3117	1.195 (0.7278–1.963) *p* = 0.5283	0.7406 (0.4056–1.352) *p* = 0.3548	0.7742 (0.4254–1.409) *p* = 0.4428	1.045 (0.5364–2.034) *p* ≥ 0.9999
	TT vs. TC+CC (Dominant)	1.682 (0.9325–3.034) *p* = 0.1020	1.217 (0.6703–2.209) *p* = 0.5462	0.5965 (0.2512–1.416) *p* = 0.2811	1.486 (0.6182–3.571) *p* = 0.5061	1.436 (0.9445–2.182) *p* = 0.1109	0.8998 (0.5932–1.365) *p* = 0.6712	1.369 (0.8187–2.288) *p* = 0.2500	1.507 (0.8948–2.539) *p* = 0.1558	1.101 (0.5990–2.024) *p* = 0.8768
	TT+TC vs. CC (Recessive)	1.918 (0.9724–3.782) *p* = 0.0646	1.167 (0.6199–2.196) *p* = 0.7477	0.6375 (0.2651–1.533) *p* = 0.3773	1.486 (0.6182–3.571) *p* = 0.5061	1.477 (0.9314–2.341) *p* = 0.1040	0.8216 (0.5200–1.298) *p* = 0.4170	1.477 (0.8485–2.570) *p* = 0.1923	1.477 (0.8485–2.570) *p* = 0.1923	1.000 (0.5406–1.850) *p* ≥ 0.9999
	TT+CC vs. TC (Over-dominant)	1.018 (0.5849–1.771) *p* ≥ 0.9999	1.053 (0.6031–1.838) *p* = 0.8876	0.9333 (0.4261–2.044) *p* ≥ 0.9999	1.000 (0.4583–2.182) *p* ≥ 0.9999	1.034 (0.6984–1.532) *p* = 0.9203	1.053 (0.7108–1.559) *p* = 0.8414	0.9941 (0.6144–1.608) *p* ≥ 0.9999	1.076 (0.6663–1.739) *p* = 0.8069	1.083 (0.6229–1.882) *p* = 0.8879
Allele	T vs. C (Major vs. minor)	0.6320 (0.4260–0.9375 *p* = 0.0278 *	0.8663 (0.5859–1.281) *p* = 0.4867	1.480 (0.8524–2.570) *p* = 0.2073	0.7302 (0.4210–1.267) *p* = 0.3270	0.7397 (0.5605–0.9761) *p* = 0.0346 *	1.126 (0.8538–1.485) *p* = 0.4379	0.7543 (0.5377–1.058) *p* = 0.1193	0.7253 (0.5170–1.018) *p* = 0.0696	0.9615 (0.6521–1.418) *p* = 0.9211

OR, odds ratio; CI, confidence intervals; N, normotensive; N^−^, normotensive HIV-negative; N^+^, normotensive HIV-positive; PE, preeclampsia; EOPE, early onset preeclampsia; EOPE^−^, early onset preeclampsia HIV-negative; EOPE^+^, early onset preeclampsia HIV-positive; LOPE, late onset preeclampsia; LOPE^−^, late onset preeclampsia HIV-negative; LOPE^+^, late onset preeclampsia HIV-positive; HIV^−^, HIV-negative; HIV^+^, HIV-positive. Asterisks (*) denote significance: * *p* < 0.05.

**Table 3 ijms-26-03271-t003:** Genotypic and allelic associations of *ADMA* gene polymorphisms *rs7521189* across pregnancy types stratified by HIV status.

SNP		N^−^ vs. PE^−^ OR (95% CI), *p* Value	N^+^ vs. PE^+^ OR (95% CI), *p* Value	EOPE^−^ vs. EOPE^+^ OR (95% CI), *p* Value	LOPE^−^ vs. LOPE^+^ OR (95% CI), *p* Value	N vs. PE OR (95% CI), *p* Value	HIV^−^ vs. HIV^+^ OR (95% CI), *p* Value	N vs. EOPE OR (95% CI), *p* Value	N vs. LOPE OR (95% CI), *p* Value	EOPE vs. LOPE OR (95% CI), p Value
*rs7521189* G>A Genotype	GG vs. AA (Co-dominant)	1.714 (0.8017–3.666) *p* = 0.1835	1.903 (0.8697–4.165) *p* = 0.1185	0.8750 (0.2818–2.717) *p* ≥ 0.9999	2.054 (0.6548–6.440) *p* = 0.2554	1.827 (1.061–3.148) *p* = 0.0398 *	1.334 (0.7791–2.286) *p* = 0.3386	1.555 (0.7904–3.058) *p* = 0.2346	2.130 (1.085–4.181) *p* = 0.0317 *	1.370 (0.6166–3.045) *p* = 0.5425
	GG vs. GA (Co-dominant)	1.943 (0.9723–3.882) *p* = 0.0828	1.418 (0.6853–2.936) *p* = 0.3640	1.167 (0.4155–3.276) *p* = 0.7977	0.9259 (0.3097–2.768) *p* ≥ 0.9999	1.669 (1.011–2.753) *p* = 0.0586	1.235 (0.7530–2.027) *p* = 0.4505	1.665 (0.9014–3.076) *p* = 0.1317	1.672 (0.8843–3.162) *p* = 0.1218	1.004 (0.4747–2.124) *p* ≥ 0.9999
	GA vs. AA (Co-dominant)	0.8824 (0.4572–1.703) *p* = 0.7394	1.342 (0.7084–2.542) *p* = 0.4189	0.7500 (0.3046–1.847) *p* = 0.6474	2.218 (0.9190–5.352) *p* = 0.0836	1.095 (0.6931–1.730) *p* = 0.7275	1.517 (0.6968–3.281) *p* = 0.3016	0.9336 (0.5326–1.637) *p* = 0.8867	1.274 (0.7375–2.200) *p* = 0.4038	1.365 (0.7321–2.543) *p* = 0.3465
	GG vs. GA+AA (Dominant)	1.851 (0.9724–3.525) *p* = 0.0766	1.595 (0.8070–3.154) *p* = 0.2287	1.050 (0.3949–2.792) *p* ≥ 0.9999	1.311 (0.4710–3.649) *p* = 0.7957	1.729 (1.084–2.759) *p* = 0.0256 *	1.273 (0.8027–2.020) *p* = 0.3484	1.623 (0.9112–2.890) *p* = 0.1243	1.847 (1.019–3.347) *p* = 0.0485 *	1.138 (0.5619–2.306) *p* = 0.8575
	GG+GA vs. AA (Recessive)	1.117 (0.6071–2.054) *p* = 0.7576	1.495 (0.8219–2.719) *p* = 0.2265	0.7829 (0.3334–1.839) *p* = 0.6654	2.171 (0.9501–4.960) *p* = 0.0988	1.298 (0.8477–1.988) *p* = 0.2357	1.159 (0.7580–1.774) *p* = 0.5171	1.106 (0.6530–1.873) *p* = 0.7868	1.511 (0.9082–2.514) *p* = 0.1150	1.366 (0.7596–2.457) *p* = 0.3711
	GG+AA vs. GA (Over-dominant)	1.457 (0.8367–2.538) *p* = 0.2061	0.9656 (0.5555–1.678) *p* ≥ 0.9999	1.264 (0.5805–2.752) *p* = 0.6922	0.5735 (0.2612–1.259) *p* = 0.2332	1.187 (0.8025–1.754) *p* = 0.4258	1.050 (0.7105–1.552) *p* = 0.8423	1.309 (0.8116–2.110) *p* = 0.2757)	1.075 (0.6663–1.736) *p* = 0.8074	0.8217 (0.4741–1.424) *p* = 0.5754
Allele	G vs. A (Major vs. minor)	1.319 (0.8930–1.949) *p* = 0.1663	1.395 (0.9410–2.068) *p* = 0.1096	0.9167 (0.5278–1.592) *p* = 0.7798	1.567 (0.8925–2.750) *p* = 0.1539	1.358 (1.030–1.792) *p* = 0.0346 *	1.161 (0.8809–1.531) *p* = 0.2918	1.242 (0.8856–1.742) *p* = 0.2289	1.487 (1.057–2.092) *p* = 0.0252 *	1.198 (0.8086–1.773) *p* = 0.4236

OR, odds ratio; CI, confidence intervals; N, normotensive; N^−^, normotensive HIV-negative; N^+^, normotensive HIV-positive; PE, preeclampsia; EOPE, early onset preeclampsia; EOPE^−^, early onset preeclampsia HIV-negative; EOPE^+^, early onset preeclampsia HIV-positive; LOPE, late onset preeclampsia; LOPE^−^, late onset preeclampsia HIV-negative; LOPE^+^, late onset preeclampsia HIV-positive; HIV-, HIV-negative; HIV+, HIV-positive. Asterisks (*) denote significance: * *p* < 0.05.

**Table 4 ijms-26-03271-t004:** Genotypic and allelic associations of *ADMA* gene polymorphism *rs805305* across pregnancy types stratified by HIV status.

SNP		N^−^ vs. PE^−^ OR (95% CI), *p* Value	N^+^ vs. PE^+^ OR (95% CI), *p* Value	EOPE^−^ vs. EOPE^+^ OR (95% CI), *p* Value	LOPE^−^ vs. LOPE^+^ OR (95% CI), *p* Value	N vs. PE OR (95% CI), *p* Value	HIV^−^ vs. HIV^+^ OR (95% CI), *p* Value	N vs. EOPE OR (95% CI), *p* Value	N vs. LOPE OR (95% CI), *p* Value	EOPE vs. LOPE OR (95% CI), *p* Value
*rs805305* C>G Genotype	CC vs. GG (Co-dominant)	2.396 (0.8613–6.668) *p* = 0.1407	0.9744 (0.3699–2.566) *p* ≥ 0.9999	0.5079 (0.1539–1.676) *p* = 0.3793	1.417 (0.3480–5.767) *p* = 0.7293	1.522 (0.7598–3.047) *p* = 0.2974	1.139 (0.5767–2.251) *p* = 0.7308	1.811 (0.8252–3.974) *p* = 0.1498	1.219 (0.5060–2.936) *p* = 0.6528	0.6730 (0.2723–1.664) *p* = 0.4983
	CC vs. CG (Co-dominant)	1.058 (0.5399–2.073) *p* ≥ 0.9999	0.7441 (0.4016–1.379) *p* = 0.3526	1.219 (0.4536–3.276) *p* = 0.8038	1.395 (0.5777–3.368) *p* = 0.5072	0.8860 (0.5638–1.393) *p* = 0.6455	1.544 (0.9793–2.436) *p* = 0.0660	0.7276 (0.4066–1.302) *p* = 0.3157	1.052 (0.6135–1.804) *p* = 0.8906	1.446 (0.7487–2.792) *p* = 0.3194
	CG vs. GG (Co-dominant)	2.265 (0.7314–7.016) *p* = 0.1789	1.310 (0.4666–3.676) *p* = 0.7931	0.4167 (0.1034–1.679) *p* = 0.3053	1.016 (0.2255–4.575) *p* ≥ 0.9999	1.717 (0.8078–3.651) *p* = 0.1874	0.7377 (0.3507–1.552) *p* = 0.4513	1. (0.2418–2.609) *p* = 0.0626	1.159 (0.4524–2.967) *p* = 0.8105	0.4655 (0.1698–1.276) *p* = 0.2070
	CC vs. CG+GG (Dominant)	1.335 (0.7368–2.419) *p* = 0.3674	0.7934 (0.4509–1.396) *p* = 0.4727	0.8635 (0.3810–1.957) *p* = 0.8353	1.400 (0.6251–3.136) *p* = 0.5393	1.020 (0.6791–1.533) *p* ≥ 0.9999	1.424 (0.9461–2.144) *p* = 0.0973	0.9565 (0.5796–1.578) *p* = 0.8991	1.087 (0.6628–1.783) *p* = 0.8003	1.137 (0.6408–2.016) *p* = 0.7703
	CC+CG vs. GG (Recessive)	2.364 (0.8610–6.489) *p* = 0.0974	1.075 (0.4174–2.770) *p* ≥ 0.9999	0.4846 (0.1503–1.563) *p* = 0.2592	1.277 (0.3224–5.059) *p* ≥ 0.9999	1.576 (0.7966–3.117) *p* = 0.2331	1.005 (0.5154–1.961) *p* ≥ 0.9999	1.973 (0.9121–4.267) *p* = 0.0982	1.200 (0.5061–2.845) *p* = 0.6595	0.6083 (0.2506–1.477) *p* = 0.3763
	CC+GG vs. CG (Over-dominant)	0.9565 (0.4931–1.856) *p* ≥ 0.9999	0.7467 (0.4087–1.364) *p* = 0.3613	1.367 (0.5191–3.598) *p* = 0.6268	1.336 (0.5631–3.171) *p* = 0.6611	0.8407 (0.5394–1.310) *p* = 0.4978	1.519 (0.9713–2.376) *p* = 0.0712	0.6713 (0.3794–1.188) *p* = 0.2089	1.029 (0.6058–1.747) *p* ≥ 0.9999	1.532 (0.8042–2.920) *p* = 0.2544
Allele	C vs. G (Major vs. minor)	1.505 (0.9244–2.450) *p* = 0.1102	0.8757 (0.5562–1.378) *p* = 0.6437	0.3804 (0.3804–1.383) *p* = 0.4126	1.320 (0.6855–2.540) *p* = 0.5063	1.131 (0.8124–1.573) *p* = 0.5009	1.263 (0.9071–1.759) *p* = 0.1784	1.162 (0.7786–1.733) *p* = 0.4710	1.100 (0.7345–1.647) *p* = 0.6782	0.9470 (0.5992–1.497) *p* = 0.9071

OR, odds ratio; CI, confidence intervals; N, normotensive; N^−^, normotensive HIV-negative; N^+^, normotensive HIV-positive; PE, preeclampsia; EOPE, early onset preeclampsia; EOPE^−^, early onset preeclampsia HIV-negative; EOPE^+^, early onset preeclampsia HIV-positive; LOPE, late onset preeclampsia; LOPE^−^, late onset preeclampsia HIV-negative; LOPE^+^, late onset preeclampsia HIV-positive; HIV-, HIV-negative; HIV+, HIV-positive.

**Table 5 ijms-26-03271-t005:** Genotypic and allelic associations of *ADMA* gene polymorphisms *rs3131383* across pregnancy types stratified by HIV status.

SNP		N^−^ vs. PE^−^ OR (95% CI), *p* Value	N^+^ vs. PE^+^ OR (95% CI), *p* Value	EOPE^−^ vs. EOPE^+^ OR (95% CI), *p* Value	LOPE^−^ vs. LOPE^+^ OR (95% CI), *p* Value	N vs. PE OR (95% CI), *p* Value	HIV^−^ vs. HIV^+^ OR (95% CI), *p* Value	N vs. EOPE OR (95% CI), *p* Value	N vs. LOPE OR (95% CI), *p* Value	EOPE vs. LOPE OR (95% CI), p Value
*rs3131383* G>T Genotype	GG vs. TT (Co-dominant)	1.841 (0.8683–3.902) *p* = 0.1306	1.250 (0.5569–2.804) *p* = 0.6828	0.4934 (0.1463–1.664) *p* = 0.3642	0.7521 (0.2842–1.991) *p* = 0.6258	1.521 (0.8785–2.634) *p* = 0.1645	0.7687 (0.4447–1.328) *p* = 0.4050	0.9354 (0.4556–1.920) *p* ≥ 0.9999	2.303 (1.223–4.337) *p* = 0.0113 *	2.462 (1.144–5.298) *p* = 0.0239 *
	GG vs. GT (Co-dominant)	2.137 (1.044–4.374) *p* = 0.0490 *	1.117 (0.5569–2.240) *p* = 0.8596	0.5921 (0.2204–1.591) *p* = 0.3269	0.8254 (0.3243–2.101) *p* = 0.8126	1.535 (0.9337–2.524) *p* = 0.1025	0.9542 (0.5828–1.562) *p* = 0.9002	1.175 (0.6374–2.167) *p* = 0.6368	2.015 (1.113–3.648) *p* = 0.0271 *	1.714 (0.8718–3.371) *p* = 0.1264
	GT vs. TT (Co-dominant)	0.8615 (0.3469–2.139) *p* = 0.8184	1.119 (0.4327–2.893) *p* ≥ 0.9999	0.8333 (0.2029–3.423) *p* ≥ 0.9999	0.9112 (0.3026–2.744) *p* ≥ 0.9999	0.9911 (0.5152–1.906) *p* ≥ 0.9999	0.8056 (0.4206–1.543) *p* = 0.6200	0.7959 (0.3402–1.862) *p* = 0.6699	1.143 (0.5457–2.394) *p* = 0.8506	1.436 (0.5933–3.475) *p* = 0.5052
	GG vs. GT+TT (Dominant)	1.993 (1.121–3.544) *p* = 0.0212 *	1.169 (0.6555–2.086) *p* = 0.6591	0.5526 (0.2400–1.273) *p* = 0.2083	0.7901 (0.3630–1.720) *p* = 0.6923	1.529 (1.018–2.297) *p* = 0.0501	0.8686 (0.5797–1.301) *p* = 0.5368	1.070 (0.6443–1.778) *p* = 0.7963	2.141 (1.313–3.490) *p* = 0.0026 **	2.000 (1.136–3.522) *p* = 0.0228 *
	GG+GT vs. TT (Recessive)	1.523 (0.7344–3.156) *p* = 0.2755	1.217 (0.5519–2.682) *p* = 0.6907	0.5585 (0.1695–1.841) *p* = 0.3848	0.8038 (0.3211–2.013) *p* = 0.8158	1.369 (0.8016–2.338) *p* = 0.2798	0.7776 (0.4562–1.326) *p* = 0.4169	0.9025 (0.4455–1.828) *p* = 0.8600	1.901 (1.036–3.490) *p* = 0.0521	2.107 (1.005–4.417) *p* = 0.0682
	GG+TT vs. GT (Over-dominant)	1.863 (0.9293–3.736) *p* = 0.0853	1.073 (0.5433–2.119) *p* = 0.8639	0.6628 (0.2516–1.746) *p* = 0.4675	0.9041 (0.3749–2.181) *p* ≥ 0.9999	1.410 (0.8687–2.289) *p* = 0.1787	1.006 (0.6223–1.627) *p* ≥ 0.9999	1.188 (0.6519–2.166) *p* = 0.6411	1.650 (0.9341–2.914) *p* = 0.0993	1.389 (0.7240–2.663) *p* = 0.4094
Allele	G vs. T (Major vs. minor)	1.743 (1.123–2.705) *p* = 0.0150 *	1.174 (0.7462–1.847) *p* = 0.4920	0.5748 (0.2965–1.114) *p* = 0.1340	0.8097 (0.4578–1.432) *p* = 0.5615	1.437 (1.049–1.969) *p* = 0.0257 *	0.8426 (0.6161–1.152) *p* = 0.3006	1.009 (0.6758–1.506) *p* ≥ 0.9999	1.449 (0.9229–2.254) *p* = 0.0004 ***	1.942 (1.260–2.994) *p* = 0.0034 **

OR, odds ratio; CI, confidence intervals; N, normotensive; N^−^, normotensive HIV-negative; N^+^, normotensive HIV-positive; PE, preeclampsia; EOPE, early onset preeclampsia; EOPE^−^, early onset preeclampsia HIV-negative; EOPE^+^, early onset preeclampsia HIV-positive; LOPE, late onset preeclampsia; LOPE^−^, late onset preeclampsia HIV-negative; LOPE^+^, late onset preeclampsia HIV-positive; HIV-, HIV-negative; HIV+, HIV-positive. Asterisks (*) denote significance: * *p* < 0.05, ** *p* < 0.01 and *** *p* < 0.001.

**Table 6 ijms-26-03271-t006:** Genetic models. Model descriptions of predisposing genotypes for the genetic models used.

Genetic Model	Description of Predisposing Genotypes
Co-dominant	Equivalence in the impact of two alleles from a gene pair.
Dominant	Alleles that display the same phenotype irrespective of the identity of the paired alleles.
Recessive	A phenotype is expressed solely when the paired alleles are identical.
Over-dominant	The heterozygote is more effective than the homozygote.

## Data Availability

Data are contained within the article and Appendix A.

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
