# Peer review of "Asymmetric Dimethylaminohydrolase Gene Polymorphisms Associated with Preeclampsia Comorbid with HIV Infection in Pregnant Women of African Ancestry"

_ijms, 2025, doi:10.3390/ijms26073271_

Round 1
Reviewer 1 Report
Comments and Suggestions for Authors
In present work, Mthembu et al. try to investigate the association of dimethylarginine dimethylaminohydrolase 1 (DDAH1) (rs669173, rs7521189) and DDAH 2 gene (rs805305, rs3131383) with the risk of preeclampsia comorbid with HIV in pregnant women of African ancestry. This study demonstrated that preeclamptic women exhibited a higher frequency of analysed variants, in contrast to those with the duality of preeclampsia and HIV infection. However, there are some questions that should be explained.
Major concerns
1. The Materials and Methods section is too simple. How to confirm the genomic DNA quality and RNA pollution? How to verify that the amplified typing results are all correct during TaqMan genotyping of ADMA gene polymorphisms?
2. There is a high similarity (45%) for this manuscript, and 23% similarity is related to the author’ previous paper (Mlambo et al. Arch Gynecol Obstet (IF 2.1). 2024). There are only tables in this manuscript. Therefore, the novelty of this manuscript is not enough for publication in IJMS.
3. There are repeats in the result section compared with Tables, which lead to a long manuscript. In addition, genotyping scatter plot should be provided.
4. English grammar and writing style should be checked and revised throughout the manuscript.
Minor concerns
1. Abstract section, a summary is needed in the end.
2. There are too many Keywords.
3. Introduction section, hypothesis and aim of this study should be included in the end of Introduction section.
4. Line 96, ‘group (<0.0001)’, ‘p’ should be added. Please check it throughout this manuscript.
5. Lines 101-102, ‘N-’, ‘N+’ ‘EOPE-’ and ‘LOPE+’, superscript should be used. Please check these throughout this manuscript。
6. Line 152, ‘p = 0.0278’, ‘p’ should be used in italic. Please check it throughout this manuscript.
7. Lines 877-878, ‘normotensive (N)’ and ‘preeclamptic (PE)’ should be presented in the first time of the text. Please check these throughout the manuscript.
8. Lines 939-945, delete ‘(IQR)’ ‘(HWE)’, ‘(OR)’, and ‘(CI)’.
9. Gene abbreviation (DDAH 1 and DDAH 2) should be in italic. Please check these throughout this manuscript.
Comments on the Quality of English LanguageThe English could be improved to more clearly express the research.
Author Response
For research article
|
Response to Reviewer 1 Comments
|
||
|
1. Summary |
|
|
|
Thank you very much for taking the time to review this manuscript. Please find the detailed responses below and the corresponding revisions/corrections highlighted in the re-submitted files. |
||
|
2. Questions for General Evaluation |
Reviewer’s Evaluation |
Response and Revisions |
|
Does the introduction provide sufficient background and include all relevant references? |
Can be improved |
|
|
Are all the cited references relevant to the research? |
Must be improved |
|
|
Is the research design appropriate? |
Must be improved |
|
|
Are the methods adequately described? |
Must be improved |
|
|
Are the results clearly presented? |
Must be improved |
|
|
Are the conclusions supported by the results? |
Must be improved |
|
|
3. Point-by-point response to Comments and Suggestions for Authors |
||
|
Comments 1: The Materials and Methods section is too simple. How to confirm the genomic DNA quality and RNA ? How to verify that the amplified typing results are all correct during TaqMan genotyping of ADMA gene polymorphisms?
|
||
|
Response 1: Thank you for this comment, we have amended the material and methods section. DNA purity was assessed by measuring the absorbance at 2260nm using the NanoDrop system. Pure DNA will have an A260/280 absorbance ratio of 1.8, and an A260/230 ratio of 2.0 or higher. Also, the Qiagen spin column-based protocols provides an efficient removal of chemical contaminants. We used specific functionally tested assays with specific VIC/FAM sequences. The changes are found on page 16, paragraph 4-5, line 923-928, 943-945.
|
||
|
Comments 2: There is a high similarity (45%) for this manuscript, and 23% similarity is related to the author’ previous paper (Mlambo et al. Arch Gynecol Obstet (IF 2.1). 2024). There are only tables in this manuscript. Therefore, the novelty of this manuscript is not enough for publication in IJMS. There are repeats in the result section compared with Tables, which lead to a long manuscript. In addition, genotyping scatter plot should be provided. |
||
|
Response 2: Thank you for noting the similarities between the two papers, however this paper showed single nucleotide polymorphisms (SNPs) related to DDAH 1 and DDAH 2 genes related to preeclampsia comorbid with HIV infection in pregnant women of African descent. Our study looked at the effect of different SNPs (rs669173, rs7521189, rs3131383 and rs805305) on ADMA levels and how it correlates with preeclampsia comorbid with HIV in pregnant women while Mlambo et al, a member of our team, investigated the association between placental growth factor and soluble FMS-like tyrosine kinase-1 in South African preeclamptic women. The similarity is in the style of presentation of the result but not the content of the results. We have, accordingly modified the results section to improve the manuscript. We have also included genotyping scatter plot in the results section. Pages 14-15, 16-24
|
||
|
4. Response to Comments on the Quality of English Language |
||
|
Point 1: English grammar and writing style should be checked and revised throughout the manuscript. |
||
|
Response 1: Thank you for the comment. We have accordingly improved the English grammar and writing style throughout the manuscript. |
||
|
5. Additional clarifications |
||
|
Abstract section, a summary is needed in the end. There are too many Keywords. Introduction section, hypothesis and aim of this study should be included in the end of Introduction section. Response: Thank you for the comment. The study aims to investigate the association of DDAH 1 (rs669173, rs7521189) and DDAH 2 gene (rs805305, rs3131383) within pregnant women with PE and HIV infection. Our study hypothesized that DDAH gene polymorphisms are significantly associated with an increased risk of PE in pregnant women of African ancestry who are also comorbid with HIV infection. These polymorphisms exacerbate endothelial dysfunction and impaired nitric oxide synthesis, intensifying the severity of PE within this population. The changes are found on page 2, paragraph 4, line 83-85, 87-91. Line 96, ‘group (<0.0001)’, ‘p’ should be added. Please check it throughout this manuscript. Response: Thank you for the comment. ‘p’ was added in line 96 and it was added where required throughout the manuscript. Lines 101-102, ‘N-’, ‘N+’ ‘EOPE-’ and ‘LOPE+’, superscript should be used. Please check these throughout this manuscript。 Response: Thank you for the comment. We have used a superscript for ‘N-’, ‘N+’, ‘EOPE-’, and ‘LOPE+’. Lines 101-102 and throughout the manuscript. Line 152, ‘p = 0.0278’, ‘p’ should be used in italic. Please check it throughout this manuscript. Responses: Thank you for the comment. In line 152, we italicized ‘p’. Lines 939-945, delete ‘(IQR)’ ‘(HWE)’, ‘(OR)’, and ‘(CI)’. Response: Thank you for the comment. In lines 939-945, we have deleted ‘(IQR)’, ‘(HWE)’, ‘(OR)’, and ‘(CI)’. Gene abbreviation (DDAH 1 and DDAH 2) should be in italic. Please check these throughout this manuscript. Response: Thank you for the comment. We have italicized (DDAH 1 and DDAH 2) throughout the manuscript.
|
||

Reviewer 2 Report
Comments and Suggestions for Authors
The paper entitled:” Asymmetric Dimethylarginine gene polymorphisms associated with preeclampsia comorbid with HIV in pregnant women of African ancestry” by Mthembu MH et al. deals with an observational study that attempts to correlate polymorphisms in the asymmetric dimethylarginine (ADMA), a naturally occurring chemical found in blood plasma, which correlates with a number of pathologies, in particular possibly also with preeclampsia in a specific population (HIV comorbid) pregnant African women, according to this study’s results.
ADMA is created during protein methylation, a common mechanism of post-translational protein modification. The methyl groups transferred to create ADMA are derived from the methyl group donor S-adenosylmethionine, an intermediate in the metabolism of homocysteine. After synthesis, ADMA migrates into the extracellular space and hence into blood plasma. The elimination of ADMA occurs through urine excretion and metabolism by the enzyme dimethylarginine dimethylaminohydrolase (DDAH).
Polymorphisms in the DDAH genes were therefore evaluated. We think the title therefore imprecise and “Asymmetric Dimethylarginine gene polymorphisms associated..” should read “dimethylarginine dimethylaminohydrolase (DDAH) gene polymorphisms associated…”
ADMA promotes vasodilatation and a spiralling effect occurs which in turn inhibits nitric oxide ( NO) production. Vasculature alterations seen in preeclampsia can be therefore rationally correlated with different ADMA concentrations, which could depend on ADMA elimination, which in turn could be tracked to DDAH activity and therefore polymorphism in their genes.
The study is interesting because it involved a relatively large number (405) of women enrolled: 204 were preeclamptic and 201 were normotensive pregnant. Whole blood was collected for DNA extraction and the genetic analysis of the DDAH 1 (rs669173, rs7521189) and DDAH 2 (rs805305, rs3131383) gene polymorphisms was performed.
Findings highlight a strong association of the rs669173 and rs7521189 single nucleotide polymorphisms (SNPs) of the DDAH 1 gene and rs3131383 of the DDAH 2 gene, indicating their role in reducing the bioavailability of NO which affects endothelial function leading to the development of preeclampsia in pregnant women of African ancestry.
To that regard, is there a known correlation with African ancestry or at least an interpretation for polymorphism association African ancestry?
In contrast, the rs805305 variant of the DDAH 2 gene was not significantly associated with preeclampsia development.
The phrase “preeclampsia is attributed to anti-HIV therapy” seems therefore unclear in this context. How “ none of the SNPs investigated correlated with HIV infection and may be attributed to the effect of antiretroviral therapy”?
Also, “there were no significant associations between HIV infection and both the DDAH 1 and DDAH 2 gene polymorphisms. Thus, HIV status did not influence the variation of these genes in pregnant women of African ancestry.”
How pregnancy or HIV can influence genotypic or allelic variations?
PLease explain that in the text
No Discussion session is present. Comments have been inserted in results. Shifting discussion in a dedicated session could help summarize and foster conclusions.
Since analytical limitations could result in missed correlation. Please discuss in relation to analytical limitations, if present. Discuss limitations of various methods used, both analytical and statistical.
In addition:
-provide a diagram of four genetic models were tested to clarify concepts in Tables 2-4
-add a diagram to exemplify results
-add some output diagram from analysis
Author Response
For review article
|
Response to Reviewer 2 Comments
|
||
|
1. Summary |
|
|
|
Thank you very much for taking the time to review this manuscript. Please find the detailed responses below and the corresponding revisions/corrections highlighted in the re-submitted files.
|
||
|
2. Questions for General Evaluation |
Reviewer’s Evaluation |
Response and Revisions |
|
Is the work a significant contribution to the field? |
|
|
|
Is the work well organized and comprehensively described? |
|
|
|
Is the work scientifically sound and not misleading? |
|
|
|
Are there appropriate and adequate references to related and previous work? |
|
|
|
Is the English used correct and readable? |
|
|
|
3. Point-by-point response to Comments and Suggestions for Authors |
ADMA is created during protein methylation, a common mechanism of post-translational protein modification. The methyl groups transferred to create ADMA are derived from the methyl group donor S-adenosylmethionine, an intermediate in the metabolism of homocysteine. After synthesis, ADMA migrates into the extracellular space and hence into blood plasma. The elimination of ADMA occurs through urine excretion and metabolism by the enzyme dimethylarginine dimethylaminohydrolase (DDAH). Polymorphisms in the DDAH genes were therefore evaluated. We think the title therefore imprecise and “Asymmetric Dimethylarginine gene polymorphisms associated..” should read “dimethylarginine dimethylaminohydrolase (DDAH) gene polymorphisms associated…”
|
Thank you for the comment. We have amended the title. |
|
Comments 1: Findings highlight a strong association of the rs669173 and rs7521189 single nucleotide polymorphisms (SNPs) of the DDAH 1 gene and rs3131383 of the DDAH 2 gene, indicating their role in reducing the bioavailability of NO which affects endothelial function leading to the development of preeclampsia in pregnant women of African ancestry. To that regard, is there a known correlation with African ancestry or at least an interpretation for polymorphism association African ancestry?
|
||
|
Response 1: Thank you for pointing this out. We have found limited data on single nucleotide polymorphisms correlated with African ancestry. Genetic variants which impair NO has been shown to increase hypertension risk (Curtis et al., 2024). In a systematic review on the association of 46 polymorphisms in 33 genes with hypertension, only rs4340, rs699 and rs5186 demonstrated a link in African populations (Yako et al., 2018). In Africa, gene-based analyses of 78 variants have shown an association of component of the renin-angiotensin aldosterone system with essential hypertension (Kalideen et al., 2024). Moreover, genetic variations on DDAH 1 genes in an Egyptian group but not in pregnancy related complications. Page 28 Paragraph 2, lines 760-766.
|
||
|
Comments 2: The phrase “preeclampsia is attributed to anti-HIV therapy” seems therefore unclear in this context. How “ none of the SNPs investigated correlated with HIV infection and may be attributed to the effect of antiretroviral therapy”? Also, “there were no significant associations between HIV infection and both the DDAH 1 and DDAH 2 gene polymorphisms. Thus, HIV status did not influence the variation of these genes in pregnant women of African ancestry.” How pregnancy or HIV can influence genotypic or allelic variations? PLease explain that in the text |
||
|
Response 2: Thank you for the comment. We apologize for the error brought about by language. We have, accordingly, revised the statements on HIV influence on genotypic or allelic variations to clarify that it is gene polymorphisms that can affect HIV infection outcomes and not HIV infection influencing genotypic variation. The modifications are in the abstract and conclusion. Pages 1 (Paragraph 1) and 17 (Paragraph 2), Line 30-31 and 914-916.
|
||
|
4. Response to Comments on the Quality of English Language |
||
|
Point 1: The quality of English does not limit my understanding of the research. |
||
|
Response 1: Thank you for the comments. |
||
|
5. Additional clarifications |
||
|
No Discussion session is present. Comments have been inserted in results. Shifting discussion in a dedicated session could help summarize and foster conclusions. Since analytical limitations could result in missed correlation. Please discuss in relation to analytical limitations, if present. Discuss limitations of various methods used, both analytical and statistical. In addition: -provide a diagram of four genetic models were tested to clarify concepts in Tables 2-4 -add a diagram to exemplify results -add some output diagram from analysis
Response: Thank you for the comments. We have added scatter plots to discuss our results to improve the understanding of our findings. Pages 10-11. Correlation may be limited by non-linear relationships and outliers in both analytical and statistical analysis. However, in this study the small sample size infers that correlation does not imply causation and the does not provide information about the magnitude of the effect. Limitations to the study under discussions are mentioned on Page 28, Paragraph 2, lines 766-774. We further included a table of the four genetic models that were tested on page 4.
|
||

Reviewer 3 Report
Comments and Suggestions for Authors
Attached

Author Response
Manuscript Critique ijms-3364767
"Asymmetric Dimethylaminohydrolase Gene Polymorphisms Associated with Preeclampsia Comorbid with HIV in Pregnant Women of African Ancestry"
- Abstract Strengths:
- The study aims to explore the association of DDAH1/DDAH2 gene polymorphisms with PE and
- This report emphasizes essential findings about genetic variations and their potential implications, making it crucial to understand their impact on health and
Weaknesses:
- There is a lack of clarity regarding the statistical methods and key quantitative
Thank you for the comment. Statistical analysis is brief due to word count. However, full statistical analysis is outlined in 4.5, Line 953 – 965.
- Phrases like "strong association" are unsupported by concrete numerical
The abstract has been amended to reflect an association and the word “strong” has been deleted. Line 25.
- Results
- Statistical Reporting
- Strengths:
- Genotype and allele frequency data are stratified by clinical subgroups (EOPE, LOPE, HIV status), enhancing the depth of the
- Chi-square tests were included to identify significant
- Weaknesses:
- Confidence intervals (CIs) and effect sizes for statistical comparisons are inconsistently reported, limiting
- Strengths:
- Statistical Reporting
We have included CIs in the text of the results and not only in the tables.
- The absence of adjustments for multiple comparisons raises concerns about Type I
Thank you for this comment. Adjustments for multiple comparisons are now shown in the results
- Subgroup Analysis
- Strengths:
- Subgroup comparisons between PE with and without HIV, as well as normotensive controls, provide valuable insights into stratified genetic
- Weaknesses:
- The biological interpretation of stratified findings (e.g., EOPE LOPE) lacks depth and clarity.
- Strengths:
Thank you for your feedback. Research indicates that EOPE and LOPE exhibit distinct epigenetic profiles, potentially elucidating the variations in outcomes. Differential variations in DNA methylation have been observed in placental and fetal tissues, potentially influencing cardiovascular and metabolic disease susceptibility in offspring; thus, we conducted this subgroup analysis (Herzorg et al., 2017). Herzog EM, Eggink AJ, Willemsen SP, Slieker RC, Wijnands KPJ, Felix JF, Chen J, Stubbs A, van der Spek PJ, van Meurs JB, Steegers-Theunissen RPM. Early- and late-onset preeclampsia and the tissue-specific epigenome of the placenta and newborn. Placenta. 2017 Oct;58:122-132. doi: 10.1016/j.placenta.2017.08.070. Epub 2017 Aug 30. PMID: 28962690.
- Data Presentation
- Strengths:
- Results are presented in an organized manner, supported by tables and
- Weaknesses:
- Scatterplots and tables are included but not thoroughly explained in the text, reducing their value for
Thank you for the comment. The scatterplots and tables have been explained in the results section.
- Discussion
- Novelty
- Strengths:
- The research focuses on a distinctive population of individuals with African ancestry, exploring the complex relationship between physical exercise (PE) and
- First, we will report on specific DDAH polymorphisms in this
- Weaknesses:
- The novelty claim is not well-supported by comparisons with existing
- Strengths:
- Novelty
I wish to respectfully indicate that lines 624-626 outlines Caucasian, Australian, Egyptian and Swedish populations that demonstrates DDAH 1 polymorphisms. Line 628-630 compares SNPs of rs7521189 in a Swedish population compared to an African population. However, the discussion has been amended to also include a paper on pulmonary hypertension where 32 Single nucleotide polymorphisms of DDAH 1 and 2 were examined in American patients with pulmonary hypertension in patients with bronchopulmonary dysplasia compared to bronchopulmonary dysplasia alone (Trittman et al., 2016). The authors report that only rs480414 of the DDAH 1 gene was protective against the development of pulmonary hypertension in patients with bronchopulmonary dysplasia (Trittman et al., 2016), Line 810-815.
- Interpretation of Results
- Strengths:
- Discusses findings in the context of endothelial dysfunction and nitric oxide bioavailability.
- Weaknesses:
- Conclusions about biological mechanisms are speculative and lack experimental
- Strengths:
Lines 772-777 outlines that genetic variants which impair NO increase hypertension risk.
- Insufficient discussion of potential confounders (e.g., antiretroviral therapy).
Future studies should incorporate routine genetic testing in all HIV patients, as this may be valuable for identifying individuals with specific SNPs that could influence their response to antiretroviral therapy. Line 827-829 and 838-840.
- Limitations
- The discussion does not adequately address the following:
- Sample size
- The discussion does not adequately address the following:
The Cohen effect calculation was done by a biostatistician. This information was included in the manuscript 4.2. Lines 916-918.
A limitation of our study is the sample size (n=405). There is a necessity for large scale analysis of different ethnic groups to accurately characterize these SNPs (Dvornyk et al., 2004).
- Potential environmental or socioeconomic
The discussion has been amended. It is true that environmental and socioeconomic confounders factors may confound the data and therefore has been included in the discussion. Line 827-830.
- Relevance of findings to broader
The relevance of our findings using an African population was compared to American, Australian, Caucasian, Swedish and Egyptian population in the discussion.
- Statistical Rigor
- Strengths:
- Appropriate use of Chi-square tests and subgroup
- Weaknesses: Lack of a priori power
A priori power analysis involves computing the sample size required to detect an effect of a given size with the desired power. With due respect, the authors would like to draw your attention to the fact that this information was outlined under 4.2, Lines 916-923.
- Missing details on whether adjustments for confounding factors were
G*power statistical analysis for the subgroup analysis in included for sample size. Adjustment for subgroup analysis has been added to the results, Lines 917-920.
- References Strengths:
- Relevant to endothelial dysfunction, SNPs, and PE
Weaknesses:
- Several self-citations and outdated
With all due respect, only two references from our laboratory were included. Also, only 5 out 98 references were before the year 2000. It is often difficult to eliminate original papers that refer to the gene.
- Limited incorporation of recent studies to strengthen the
Current references have been added to strengthen the discussion.
- Strengths and Weaknesses Strengths:
- Novel focus on African populations and the combined impact of PE and
- Comprehensive genotype analysis with stratified subgroup
- Potential contribution to understanding endothelial dysfunction in
Weaknesses:
- Insufficient statistical detail (e.g., effect sizes, CIs, and corrections for multiple testing).
Sample size and effect size was outlined in 4.2. The CIs and adjusted p – values for multiple testing is now included in the results.
- Speculative biological interpretations without experimental
- Limited discussion of limitations and broader
Limitations have been added in line 827-832.
- Ethical transparency on data availability and approvals is
The original manuscript includes institutional ethical approval was outlined in 4.1, line 885-887.
- Recommendation Decision: Major Revision
- Statistical reporting must be improved by adding power analysis, effect sizes, and proper corrections.
- The biological relevance of findings must be clarified and supported with references or evidence.
- The discussion should include a balanced analysis of limitations and consider broader
- References must be updated, avoiding over-reliance on self-citations.
Major revisions to statistics, relevance, references, limitations have been completed and highlighted in red in the manuscript. Additionally, we have improved the conclusion to better understand the effect of SNPs on the downstream effect of endothelial impairment in preeclampsia.

Reviewer 4 Report
Comments and Suggestions for Authors
Thank you for the opportunity to review this paper. The authors interpret different patterns of SNPs in African pregnant women diagnosed with preeclampsia compared to controls. However, the frequencies of candidate mutations in SNPs do not exhibit disparities in HIV + and - individuals.
I want to recommend the MS for publication but after some minor modifications.
In the abstract section:
The names of the DDAH1 and DDAH2 mutations are enlisted three times. I think the names of the SNPs have to be explained once or twice in the abstract and the materials and methods section of the abstract can be more concise. The number of recruited HIV women is omitted in the Materials and Method section.
The abstract results go beyond the presented results in the MS. The authors explain the different levels of NO as the consequences of the variations of the SNP frequencies; however, they did not measure the NO activity/levels (in lines 25-28).
Line 30 instead of "may" it sounds more adequate if the authors change it to "can"
In the Introduction:
In line 66, the authors should write out NOS to Nitric oxide synthetase
In line 69, the reference is left out, and the authors must provide a reference for the recently inserted sentence.
In line 78: typographic error "African descent." to "African descent"
A sentence in 83-85 may be beyond the scope and not an adequate statement. The authors did not explore pulmonary embolism in pregnancies complicated with HIV in the present MS.
Results:
The authors had a robust number of subjects that may allow the significance of calculation between SNPs and PE and even HIV status.
The presentation of sociodemographic and clinical data is a little bit unclear. The data regarding the comparisons of HIV to non-HIV counterparts are not presented (i.e. N+ vs N- and so on). From the data can man conclude that most obviously there were no differences, but then the authors can declare this statement in one or two sentences.
Another remark is that the authors could even examine their results with multiple analyses (maybe the maternal weight/age may influence their results on PE).
2.2 point is too verbose, and it should be abridged because it contains textbook facts.
2.3 results are very long and a direct repetition of table 3. Maybe some of the results can be excluded. I think that table 3 should go to the end of the MS.
Materials and methods
In line 853, ART has no meaning
In line 857, ARV must be explained.
In the Discussion section, no remarks were found.
Author Response
|
1. Summary |
|
|
|
Thank you very much for taking the time to review this manuscript. Please find the detailed responses below and the corresponding revisions/corrections highlighted/in track changes in the re-submitted files.
|
||
|
2. Questions for General Evaluation |
Reviewer’s Evaluation |
Response and Revisions |
|
Does the introduction provide sufficient background and include all relevant references? |
Yes |
|
|
Are all the cited references relevant to the research? |
Yes |
|
|
Is the research design appropriate? |
Yes |
|
|
Are the methods adequately described? |
Can be improved |
|
|
Are the results clearly presented? |
Yes |
|
|
Are the conclusions supported by the results? |
Yes |
|
|
3. Point-by-point response to Comments and Suggestions for Authors Thank you for the opportunity to review this paper. The authors interpret different patterns of SNPs in African pregnant women diagnosed with preeclampsia compared to controls. However, the frequencies of candidate mutations in SNPs do not exhibit disparities in HIV + and - individuals. I want to recommend the MS for publication but after some minor modifications. |
||
|
Comments 1: In the abstract section: The names of the DDAH1 and DDAH2 mutations are enlisted three times. I think the names of the SNPs have to be explained once or twice in the abstract and the materials and methods section of the abstract can be more concise. The number of recruited HIV women is omitted in the Materials and Method section.
|
||
|
Response 1: Thank you for pointing this out. We have removed some names of the SNPs and added the number of HIV women recruited for the study. |
||
|
Comments 2: The abstract results go beyond the presented results in the MS. The authors explain the different levels of NO as the consequences of the variations of the SNP frequencies; however, they did not measure the NO activity/levels (in lines 25-28). |
||
|
Response 2: I appreciate your feedback. We have duly cited the source of the information utilized. |
||
|
4. Response to Comments on the Quality of English Language |
||
|
Point 1: |
||
|
Response 1: |
||
|
5. Additional clarifications |
||
|
Line 30 instead of "may" it sounds more adequate if the authors change it to "can" Thank you, we have replaced “may” with “can”.
In the Introduction:
In line 66, the authors should write out NOS to Nitric oxide synthetase Nitric oxide synthase has been abbreviated in line 58 to NOS and then maintained throughout the manuscript.
In line 69, the reference is left out, and the authors must provide a reference for the recently inserted sentence. Thank you for the comment. We have added the reference.
In line 78: typographic error "African descent." to "African descent" There was no error in line 78.
A sentence in 83-85 may be beyond the scope and not an adequate statement. The authors did not explore pulmonary embolism in pregnancies complicated with HIV in the present MS. Thank you for the comment. We agree, there was an error. We meant preeclampsia.
Results:
The authors had a robust number of subjects that may allow the significance of calculation between SNPs and PE and even HIV status.
The presentation of sociodemographic and clinical data is a little bit unclear. The data regarding the comparisons of HIV to non-HIV counterparts are not presented (i.e. N+ vs N- and so on). From the data can man conclude that most obviously there were no differences, but then the authors can declare this statement in one or two sentences. Thank you for this comment. The results have been amended to include a statement outlining non-significant differences based on HIV status. Line 107-108
Another remark is that the authors could even examine their results with multiple analyses (maybe the maternal weight/age may influence their results on PE). Thank you for this comment. Statistical differences between groups for maternal weight is shown in table 1 and line 104-106 whilst maternal age is outlined in line 108-109.
2.2 point is too verbose, and it should be abridged because it contains textbook facts. Thank you for the comment. We have summarized the paragraph 2.2
2.3 results are very long and a direct repetition of table 3. Maybe some of the results can be excluded. I think that table 3 should go to the end of the MS. Thank you for the comment. The table has been moved to the end as Supplementary Table 1, page 35.
Materials and methods
In line 853, ART has no meaning ART has been written in full.
In line 857, ARV must be explained. ARV has been written in full. |
||

Round 2
Reviewer 1 Report
Comments and Suggestions for Authors
Thanks for author’s responses. However, there are some questions that should be explained.
1. There is still a high similarity (40%) for this manuscript, and 23% similarity is related to the authors’ previous paper (Mlambo et al. Arch Gynecol Obstet (IF 2.1). 2024). I do not think this manuscript is suitable for publication in IJMS. Arch Gynecol Obstet (IF 2.1) may be suitable.
2. Lines 870-871, ‘Pure DNA will have an A260/280 absorbance ratio of 1.8, and an A260/230 ratio of 2.0 or higher’. I want to know how many A260/280 absorbance ratio and A260/230 ratio are in this study. Have authors performed these?
3. English grammar and writing style still should be checked and revised throughout the manuscript.
Line 5, ‘Samukelisiwe Sibiya 3 Zinhle Pretty Mlambo’. A comma is lost.
Abstract section, delete ‘(N)’ and ‘(SNPs)’.
Line 25, ‘preeclampsia’ to PE.
I have no more time to check English grammar and writing style throughout the manuscript.
4. Figures 1, 2, 3, and 4 should be merged into one Figure, and figure legend is needed.
5. Figure 5 is in low quality. It should be colorful.
6. The format of reference is not suitable for this Journal. Names of Journal should be abbreviated.
Comments on the Quality of English LanguageThe English could be improved to more clearly express the research.
Author Response
|
1. Summary |
|
|
|
Thank you very much for taking the time to review this manuscript. Please find the detailed responses below and the corresponding revisions/corrections highlighted/in track changes in the re-submitted files.
|
||
|
2. Questions for General Evaluation |
Reviewer’s Evaluation |
Response and Revisions |
|
Does the introduction provide sufficient background and include all relevant references? |
Must be improved |
|
|
Are all the cited references relevant to the research? |
Must be improved |
|
|
Is the research design appropriate? |
Must be improved |
|
|
Are the methods adequately described? |
Must be improved |
|
|
Are the results clearly presented? |
Must be improved |
|
|
Are the conclusions supported by the results? |
Must be improved |
|
|
3. Point-by-point response to Comments and Suggestions for Authors |
||
|
Comments 1: There is still a high similarity (40%) for this manuscript, and 23% similarity is related to the authors’ previous paper (Mlambo et al. Arch Gynecol Obstet (IF 2.1). 2024). I do not think this manuscript is suitable for publication in IJMS. Arch Gynecol Obstet (IF 2.1) may be suitable. |
||
|
Response 1: I would like to respectfully inform you that the manuscript has undergone a full reduction in similarity index post the Turnitin report. |
||
|
Comments 2: Lines 870-871, ‘Pure DNA will have an A260/280 absorbance ratio of 1.8, and an A260/230 ratio of 2.0 or higher’. I want to know how many A260/280 absorbance ratio and A260/230 ratio are in this study. Have authors performed these? |
||
|
Response 2: I apologize for this error, the ratio of absorbance at 260/280 nm was used to assess DNA purity was used for all samples. A ratio of approximately 1.8 is accepted as pure DNA. |
||
|
4. Response to Comments on the Quality of English Language |
||
|
Point 1: English grammar and writing style still should be checked and revised throughout the manuscript. |
||
|
Response 1: Thank you for the come, we have checked and revised English grammar and writing style throughout the manuscript. |
||
|
5. Additional clarifications |
||
|
|
||
Line 5, ‘Samukelisiwe Sibiya 3 Zinhle Pretty Mlambo’. A comma is lost.
Thank you, we have added the lost comma.
Abstract section, delete ‘(N)’ and ‘(SNPs)’.
(N) and (SNPs) were deleted in the abstract.
Line 25, ‘preeclampsia’ to PE.
In line 25, we have replaced ‘preeclampsia’ with PE.
I have no more time to check English grammar and writing style throughout the manuscript.
- Figures 1, 2, 3, and 4 should be merged into one Figure, and figure legend is needed.
Figure 1,2,3, and 4 has been merged into one Figure (Figure 1).
- Figure 5 is in low quality. It should be colorful.
Figure 5 has been improved and changed to Figure 2.
- The format of reference is not suitable for this Journal. Names of Journal should be abbreviated.
The names of journals have been abbreviated.

Reviewer 2 Report
Comments and Suggestions for Authors
In the revised version of the manuscript our previous concerns have been convincingly addressed.
Author Response
Thank you for your feedback.
Round 3
Reviewer 1 Report
Comments and Suggestions for Authors
Thanks for author’s responses. It is the third for me to review this manuscript. However, there are STILL some previous questions that should be explained. We do not think that this manuscript is suitable for publication in IJMS at present.
1. English grammar and writing style STILL should be checked and revised throughout the manuscript. I suggest you to seek the support of a Professional English proofreading and editing service.
Line 5, ‘Samukelisiwe Sibiya, 3 Zinhle Pretty Mlambo’. The comma is not in the suitable site.
Figure 1, figure legend is needed.
2. The format of reference is STILL not suitable for this Journal.
Comments on the Quality of English LanguageThe English could be improved to more clearly express the research.
Author Response
For research article
|
Response to Reviewer X Comments
|
||
|
1. Summary |
|
|
|
Thank you very much for taking the time to review this manuscript. Please find the detailed responses below and the corresponding revisions/corrections highlighted/in track changes in the re-submitted files.
|
||
|
2. Questions for General Evaluation |
Reviewer’s Evaluation |
Response and Revisions |
|
Does the introduction provide sufficient background and include all relevant references? |
Must be improved |
|
|
Are all the cited references relevant to the research? |
Must be improved |
|
|
Is the research design appropriate? |
Must be improved |
|
|
Are the methods adequately described? |
Must be improved |
|
|
Are the results clearly presented? |
Must be improved |
|
|
Are the conclusions supported by the results? |
Must be improved |
|
|
3. Point-by-point response to Comments and Suggestions for Authors |
||
|
Comments 1: Thanks for author’s responses. It is the third for me to review this manuscript. However, there are STILL some previous questions that should be explained. We do not think that this manuscript is suitable for publication in IJMS at present. English grammar and writing style STILL should be checked and revised throughout the manuscript. I suggest you to seek the support of a Professional English proofreading and editing service.
Line 5, ‘Samukelisiwe Sibiya, 3 Zinhle Pretty Mlambo’. The comma is not in the suitable site.
Figure 1, figure legend is needed.
|
||
|
Response 1: Thank you for the feedback. The manuscript has undergone Professional English proofreading and editing by a trusted English grammar platform. A report has been attached to this document.
|
||
|
Thank you for the comment. We have added the comma at the suitable site.
Thank you for the comment, we have added a figure legend to Figure 1.
Comments 2: The format of reference is STILL not suitable for this Journal. |
||
|
Response 2: Thank you for the comment. We have improved the references according to the journal’s guidelines. |
||
|
4. Response to Comments on the Quality of English Language |
||
|
Point 1: The English could be improved to more clearly express the research. |
||
|
Response 1: Thank you for the comment. We have checked and revised English grammar and writing style throughout the manuscript. |
||
|
|
||
|
|
||
Untitled
by Mbuso Mthembu
General metrics
69,929 10,596 429 42 min 23 sec 1 hr 21 min
characters words sentences reading
time
speaking time
Score Writing Issues
|
98 |
104
Issues left Critical
104
Advanced
This text scores better than 98% of all texts checked by Grammarly
Unique Words 11%
Measures vocabulary diversity by calculating the percentage of words used only once in your document
unique words
Rare Words 49%
Measures depth of vocabulary by identifying words that are not among the 5,000 most common English words.
rare words
Word Length 4.9
Measures average word length characters per word
Sentence Length 24.7
Measures average sentence length words per sentence

Round 4
Reviewer 1 Report
Comments and Suggestions for Authors
Thanks for author’s responses. It is the fourth time for me to review this manuscript. However, the writing style STILL should be checked and revised throughout the manuscript. I suggest you to seek the support of a Professional English proofreading and editing service.
For example, the abbreviation for gene should be in italic, which should be checked throughout the manuscript.
Lines 122-125, in the result section, there should not include discussion and reference.
The format of reference is TILL should be corrected. Some references include DOI (Ref.7, 70), but others are not.
Author Response
|
Response to Reviewer X Comments
|
||
|
1. Summary |
|
|
|
Thank you very much for taking the time to review this manuscript. Please find the detailed responses below and the corresponding revisions/corrections highlighted/in track changes in the re-submitted files.
|
||
|
2. Questions for General Evaluation |
Reviewer’s Evaluation |
Response and Revisions |
|
Does the introduction provide sufficient background and include all relevant references? |
Can be improved |
|
|
Are all the cited references relevant to the research? |
Can be improved |
|
|
Is the research design appropriate? |
Can be improved |
|
|
Are the methods adequately described? |
Can be improved |
|
|
Are the results clearly presented? |
Must be improved |
|
|
Are the conclusions supported by the results? |
Can be improved |
|
|
3. Point-by-point response to Comments and Suggestions for Authors |
||
|
Comments 1: Thanks for author’s responses. It is the fourth time for me to review this manuscript. However, the writing style STILL should be checked and revised throughout the manuscript. I suggest you to seek the support of a Professional English proofreading and editing service.
For example, the abbreviation for gene should be in italic, which should be checked throughout the manuscript. |
||
|
Response 1: Thank you for the comment. The manuscript has been submitted to a Professional English proofreader and editor. We have thoroughly worked on the writing style and the grammatical errors. We have ensured that we write the abbreviated genes in italics (highlighted in red) throughout the manuscript. |
||
|
Comments 2: Lines 122-125, in the result section, there should not include discussion and reference |
||
|
Response 2: Thank you for the comment. We have moved the section from lines 122-125 to the Materials and Methods section because we think it suits us better. It is now in Line 911-922. Comments 3: The format of reference STILL should be corrected. Some references include DOI (Ref.7, 70), but others are not. Response 3: Thank you for the comment. We have updated the references (7, 70) and maintained a standard references list. The DOI number has been removed from all the references. |
||
|
4. Response to Comments on the Quality of English Language |
||
|
Point 1: The English is fine and does not require any improvement. |
||
|
Response 1: Thank you for the feedback. |
||
|
|
||
|
|
||
